# EcomEval: Towards Reliable Evaluation of Large Language Models for Multilingual and Multimodal E-Commerce Applications

## Abstract

Large Language Models (LLMs) excel on general-purpose NLP benchmarks, yet their capabilities in specialized domains remain underexplored. In e-commerce, existing evaluations—such as EcomInstruct, ChineseEcomQA, eCeLLM, and Shopping MMLU—suffer from limited task diversity (e.g., lacking product guidance and after-sales issues), limited task modalities (e.g., absence of multimodal data), synthetic or curated data, and a narrow focus on English and Chinese, leaving practitioners without reliable tools to assess models on complex, real-world shopping scenarios. We introduce EcomEval, a comprehensive multilingual and multimodal benchmark for evaluating LLMs in e-commerce. EcomEval covers six categories and 37 tasks (including 8 multimodal tasks), sourced primarily from authentic customer queries and transaction logs, reflecting the noisy and heterogeneous nature of real business interactions. To ensure both quality and scalability of reference answers, we adopt a semi-automatic pipeline in which large models draft candidate responses subsequently reviewed and modified by over 50 expert annotators with strong e-commerce and multilingual expertise. We define difficulty levels for each question and task category by averaging evaluation scores across models with different sizes and capabilities, enabling challenge-oriented and fine-grained assessment. EcomEval also spans eight languages—including four low-resource Southeast Asian languages—offering a multilingual perspective absent from prior work. We evaluate 19 open and proprietary LLMs on EcomEval, revealing substantial performance disparities and highlighting scenarios where these general-purpose models perform poorly in the e-commerce domain. By combining diversity, authenticity, quality, difficulty awareness, multilinguality and multimodality, EcomEval establishes a rigorous and representative testbed for advancing research and deployment of LLMs in e-commerce. Upon acceptance, we will release the full dataset to support reproducible research.

## 1 Introduction

Large Language Models (LLMs) have achieved remarkable performance across general NLP benchmarks, but evaluating them in specialized domains like e-commerce remains challenging. Numerous benchmarks target broad capabilities – from knowledge-intensive QA to multi-task reasoning – providing ample evaluation data in generic settings. However, in the e-commerce domain, such dedicated benchmarks are sparse (Chen et al., 2025; Jin et al., 2024). This scarcity leaves a gap in assessing whether advanced LLMs possess the nuanced skills required for e-commerce applications. Practitioners currently lack reliable yardsticks to measure model competence on complex shopping-related queries, making it difficult to gauge how well these models handle real-world e-commerce tasks.

Recent efforts have started to explore e-commerce-specific LLM benchmarks, but each comes with limitations. Notable examples include EcomInstruct (Li et al., 2023), ChineseEcomQA (Chen et al., 2025), eCeLLM (Peng et al., 2024), and Shopping MMLU (Jin et al., 2024). EcomInstruct and eCeLLM introduced large instruction-tuned datasets for e-commerce, covering tasks like product description generation, attribute extraction, and recommendation. However, these rely heavily on synthetic or curated instruction data, which may not capture the full complexity of real user inter-

actions. ChineseEcomQA focuses on fundamental concept question-answering in the e-commerce domain, but it is restricted to QA pairs (primarily in Chinese) and does not encompass the rich variety of e-commerce tasks. Shopping MMLU provides a broader multi-task evaluation with 57 tasks derived from Amazon data, covering areas such as concept understanding and knowledge reasoning. Yet, its format (largely multiple-choice questions) simplifies many challenges, and like the others, it overrepresents English and Chinese content. Common shortcomings across these benchmarks include limited task diversity and a reliance on instruction-tuned or synthetic data. Important real-world scenarios – for example, interactive product guidance dialogues or after-sales customer service queries – remain largely absent from existing evaluations. These issues underscore the need for a more comprehensive benchmark, one that spans diverse tasks, languages, modalities and realistic e-commerce complexities. Our work aims to fill this critical gap.

In this paper, we present ECOMEVAL, a new benchmark that delivers **a comprehensive and realistic evaluation suite for LLMs in the e-commerce domain**. First, it achieves **broad task diversity and multimodality**, encompassing six primary categories and 37 distinct tasks that reflect genuine business needs, ranging from customer product question answering and attribute reasoning to intent understanding from queries or reviews and multi-turn shopping-assistant dialogues. Second, the benchmark is grounded in **authentic data**: most items are derived from real user queries and transaction logs, rather than purely synthetic or instruction-tuned sources, thereby capturing the complexity and noise inherent in real customer–merchant interactions. Third, it ensures **high quality and scalability** through a semi-automatic construction pipeline in which large language models generate initial drafts that are subsequently refined by over 50 expert annotators with strong e-commerce and multilingual expertise, balancing fast production with rigorous quality control. Fourth, the benchmark is explicitly **challenge-oriented**, incorporating calibrated difficulty levels at both task and item granularity, enabling fine-grained discrimination of model capabilities and surfacing performance gaps across systems. Finally, it supports **multilingual evaluation**, covering eight languages—including English, Chinese, Spanish and five additional Southeast Asian languages—thus moving beyond the English/Chinese focus of prior work and reflecting the truly global nature of e-commerce. By integrating diverse task coverage, authentic data, quality-assured scalability, difficulty-aware design, and multilingual breadth, our benchmark offers a rigorous and representative testbed for advancing LLM research and deployment in e-commerce.

We validate the effectiveness of our benchmark through extensive experiments. A total of 19 state-of-the-art LLMs – both open-source models and commercial systems – were evaluated on the full range of tasks. The results demonstrate that our benchmark robustly differentiates model capabilities, revealing significant performance disparities that were obscured by earlier, narrower evaluations. No single model dominated across all tasks, and even generally strong LLMs struggled with certain e-commerce challenges. For instance, models with impressive general NLP performance often faltered on tasks requiring nuanced understanding of domain-specific terminology or multi-step reasoning based on product knowledge. In particular, we observed that some top-performing general models (e.g., GPT-4o-tier systems) underperform on complex tasks like cross-lingual product QA and conversational recommendation, highlighting clear performance gaps. These findings are in line with recent observations that domain-tuned models can surpass even the best general LLMs on specialized e-commerce tasks. By exposing where current models excel and where they fail, the benchmark provides actionable insights into the strengths and weaknesses of today's LLMs in e-commerce. This differentiation underlines the benchmark's value in driving progress: it pinpoints which e-commerce scenarios remain most challenging for LLMs, guiding researchers and practitioners toward targeted improvements.

**Main Contributions.** We advance LLM evaluation for e-commerce through three key contributions:

- **Comprehensive, Authentic, Multilingual and Multimodal E-commerce Classification System:** We introduce a comprehensive classification system for e-commerce tasks, covering six categories and 37 diverse tasks (including 8 multimodal tasks), built from real e-commerce queries and transaction logs. It spans eight languages (English, Chinese, Spanish, Indonesian, Vietnamese, Thai, Malay, Portuguese), addressing low-resource settings and reflecting the global breadth of online e-commerce.
- **Open, High-Quality, Difficulty-Aware Dataset:** We publicly release ECOMEVAL ($\approx$ 7,200 items, including both single-turn and multi-turn questions) with reference answers and cali-

brated difficulty levels at both task and item granularity, enabling reproducible research and rigorous yet accessible evaluation.

- **Extensive Evaluation and Insights:** We benchmark 19 state-of-the-art LLMs on ECOMEVAL, revealing clear strengths and weaknesses across tasks and offering actionable guidance for future model development and deployment in e-commerce.

## 2 RELATED WORK

### 2.1 LLMS FOR E-COMMERCE

LLMs show great potential in improving day-to-day online shopping experience (Palen-Michel et al., 2024). For example, LLMs are used in product attribute extraction (Fang et al., 2024; Zhang et al., 2025), user query understanding for better search relevance (Wang & Na, 2024; Tang et al., 2025), and personalised product recommendations (Wang et al., 2024; Xu et al., 2024). To achieve good performance in e-commerce applications, the first e-commerce instruction dataset, *EcomInstruct* (Li et al., 2023), is proposed with instruction data from various e-commerce tasks. *ECInstruct* (Peng et al., 2024) covers a wider range of tasks, with each instruction data consisting of an instruction, an input, and an output.

### 2.2 E-COMMERCE BENCHMARKS

In Table 1, we compare EcomEval with related e-commerce datasets. "Difficulty-aware" indicates whether tasks are stratified by difficulty level,"Multi-cls" denotes whether tasks are categorized into multiple fine-grained classes,and "Real-world" signifies whether these datasets' tasks are mainly from the real world. EcomInstruct-test(Li et al., 2023) primarily draws from open-source datasets. eCeLLM (ECInstruct-test)(Peng et al., 2024) is derived entirely from real-world scenarios, yet it lacks multimodal tasks which are critically important in e-commerce applications. MMECInstruct-test(Ling et al., 2024) addresses this gap. To address the absence of comprehensive evaluation resources for Chinese e-commerce ecosystems, ChineseEcomQA (Chen et al., 2025), a question-answering benchmark is proposed. However, most of the aforementioned evaluation datasets suffer from limited language coverage and overly homogeneous task taxonomies. To benchmark the reasoning and multi-lingual abilities of e-commerce LLMs, Shopping MMLU (Jin et al., 2024) is proposed with tasks covering categories in shopping concept understanding, shopping knowledge reasoning, and user behavior alignment. Nevertheless, Shopping MMLU contains partially synthetic conversation data and lacks multimodal task support. Moreover, none of these datasets incorporate difficulty-tiered task design — a crucial component for fine-grained model evaluation.

Table 1: EcomEval vs. Peer E-Commerce Datasets.

| Dataset | Languages | Tasks | Multimodal | Difficulty-aware | Multi-cls | Real-world |
|---|---|---|---|---|---|---|
| ECInstruct-test | 1 | 10 | ✗ | ✗ | ✓ | ✓ |
| MMECInstruct-test | 1 | 7 | ✓ | ✗ | ✗ | ✓ |
| ChineseEcomQA | 1 | 10 | ✗ | ✗ | ✗ | ✗ |
| EcomInstruct-test | 2 | 12 | ✗ | ✗ | ✗ | ✗ |
| Shopping MMLU | 6 | 57 | ✗ | ✗ | ✓ | ✓ |
| **EcomEval(Ours)** | 7 | 37 | ✓ | ✓ | ✓ | ✓ |

## 3 ECOMEVAL

In this section, we first introduce the classification logic of the EcomEval task taxonomy. We elaborate on the six primary categories along with their corresponding specific tasks. Then, we describe the definition of the difficulty level. Finally, we provide a detailed explanation of the dataset construction methodology.

## 3.1 INTRODUCTION TO ECOMEVAL TASK TREE

Our e-commerce task tree is hierarchical. Its design philosophy is to cover as many real-world e-commerce scenarios as possible. Specifically, our task tree includes 6 primary categories and dozens of tasks. All tasks under different categories are non-overlapping. The e-commerce tasks in EcomEval mainly come from our internal application scenarios. The rest of the benchmark originates from open-source e-commerce data for which we have provided Southeast Asian multilingual versions through our internal annotation team. The double-layered donut chart in Figure 1 illustrates our e-commerce tasks, with the inner ring representing the six main categories and the outer ring showing the specific tasks.

The six primary categories are: Ecom Services, Product Understanding, User Query&Review, Shopping Reasoning, Ecom Generation and Ecom Multimodal. Among these, Ecom Services mainly includes after-sales and order related services. Product Understanding focuses on the product attributes, features, classifications, etc. User Query&Review mainly refers to the comprehension of user behavior in the e-commerce domain. Shopping Reasoning primarily involves tasks such as product recommendation, price calculation, and other reasoning-related activities. Ecom Generation, mainly encompasses content generation related to product information and descriptions. The last category, Ecom Multimodal, tasks under this category combine multimodal and textual information, including product cover image understanding, prohibited product image detection, etc. Additionally, the dataset includes eight languages: Chinese, English, Spanish, Vietnamese, Thai, Indonesian, Malay, and Portuguese, addressing the lack of low-resource language data in conventional e-commerce datasets. The details of some tasks' example can be found in the Appendix B.

In addition, to make the task tree more comprehensive, we sample data from some tasks in the eCeLLM and Shopping MMLU datasets, and translate these data from English into low-resource languages. As demonstrated in Section 2.2, the data utilized by ECeLLM are entirely derived from real-world sources. Specifically, we selectively extract those subsets from the Shopping MMLU benchmark that contain authentic product information. These measures are implemented to rigorously ensure the fidelity of the EcomEval dataset. The tasks of product inquiry, product recommendation, similar product identification, review sentiment classification, and product attribute extraction are jointly derived from eCeLLM and Shopping MMLU. Meanwhile, the numerical reasoning subset is exclusively drawn from the Shopping MMLU benchmark, while the review sentiment classification subset is uniquely derived from the eCeLLM dataset.

## 3.2 DIFFICULTY LEVEL

Our dataset's second-tier tasks come from authentic e-commerce scenarios and encompasse a diverse array of question formats, ranging from multiple-choice questions to open-ended, generation-based tasks. The difficulty levels of the questions vary accordingly. Following the scoring rubric outlined in Appendix C, we evaluate each response from the tested models shown in section 4.2 (ranging from 7B to several hundred billion parameters)on a 0–3 point scale. For any given model, we first compute the average score across all instances within each second-tier task category. Then, we compute per-task averages across all models to derive the final average score. We linearly rescale the average score to a 0–100 point range for easier interpretation and comparison. Based on the average scores of these tasks, we categorize the tasks into three difficulty levels: easy, medium, and hard. Specifically, tasks with an average score below 70 are labeled as hard, those with scores between 70 and 80 are classified as medium, and those exceeding 80 are considered easy. Appendix D provides examples of tasks across different difficulty levels. EcomEval benchmark strategically controls the distribution of task difficulty levels. Specifically, the dataset comprises 20% hard instances , 50% medium difficulty instances, and 30% easy instances.

## 3.3 DATASET CONSTRUCTION METHODOLOGY

As shown in Figure 2, this paper constructs EcomEval through the following four steps.

**Step 1: Online Log Collection.** We have collected online logs when users used open-source or closed-source large language models. These logs are categorized into two types, one type is API invoke data, where users solve specific business problems by calling LLM APIs. During a single usage session, users often submit multiple requests related to a particular business scenario, which

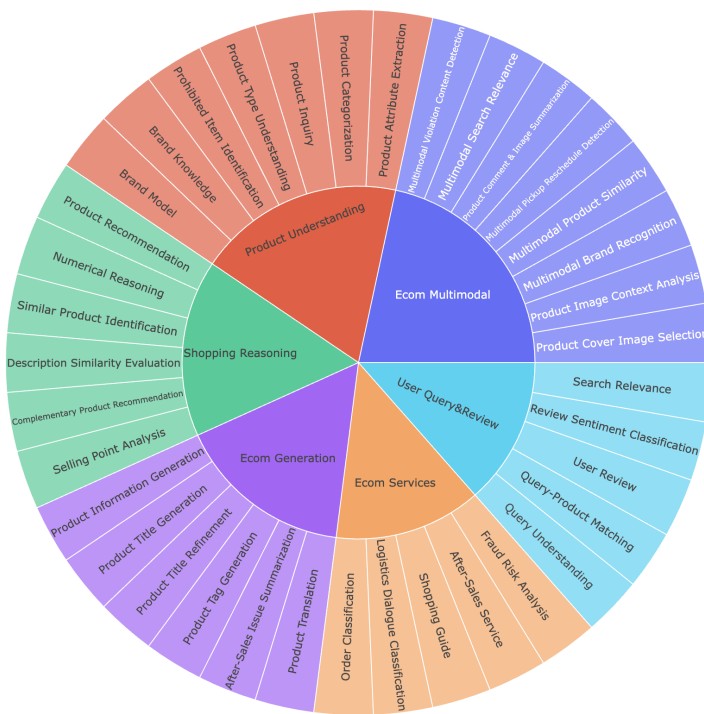

Figure 1: Overview of the EcomEval benchmark, organized into six primary categories—Ecom Services, Product Understanding, User Query&Review, Shopping Reasoning, Ecom Generation, and Ecom Multimodal—covering both user- and merchant-oriented tasks. The dataset spans eight languages (Chinese, English, Spanish, Vietnamese, Thai, Indonesian, Malay, and Portuguese), addressing the scarcity of multilingual resources in e-commerce evaluation.

tend to have similar or even identical prompts. This means that API business data for the same type of task contains different inputs but shares similar instruction contexts. The other type is data from LLM websites. Users request LLM services through a chat interface, and the prompts in this type of data are relatively random.

**Step 2: Task Classification.** API call queries within the same business usually share the same prompt instruction prefix. Therefore, for such queries, we adopt a prefix clustering method to extract different tasks. Assume that each task requires at least $n$ samples (in this work, $n = 1000$). We take the first $m$ (we try $m = 10, 20, 50$ to extract different tasks) characters from the API call query to obtain *prefix_m*, then classify the logs based on *prefix_m*. We assume that questions sharing the same *prefix_m* belong to the same task and are therefore grouped together accordingly. For clusters containing a large number of queries, we increase $m$ to perform more fine-grained clustering within that cluster.

We obtain several clusters through the prefix clustering method. Then, we employ GPT-4o to automatically generate a task name of approximately 10 words, which is then manually verified for accuracy. This method works well for API data with identical prompts. However, for website data, it is often difficult to classify using prefixes. Therefore, we fine-tune a classification model to categorize website data.

We construct the training dataset for the task classification model based on our internal e-commerce scenarios. To ensure semantic accuracy and annotation quality, the raw corpus is meticulously labeled through manual annotation. The design of the classification model's label integrates clustering analysis results from historical API logs of the platform, along with task type structures from several mainstream open-source e-commerce datasets, ultimately establishing 37 high-frequency and business-representative e-commerce task categories, as shown in Figure 1. Each task category contains 1,000 annotated samples that have undergone cleaning, deduplication, and standardization, ensuring balanced data distribution across all categories. To further enhance the model's adaptabil-

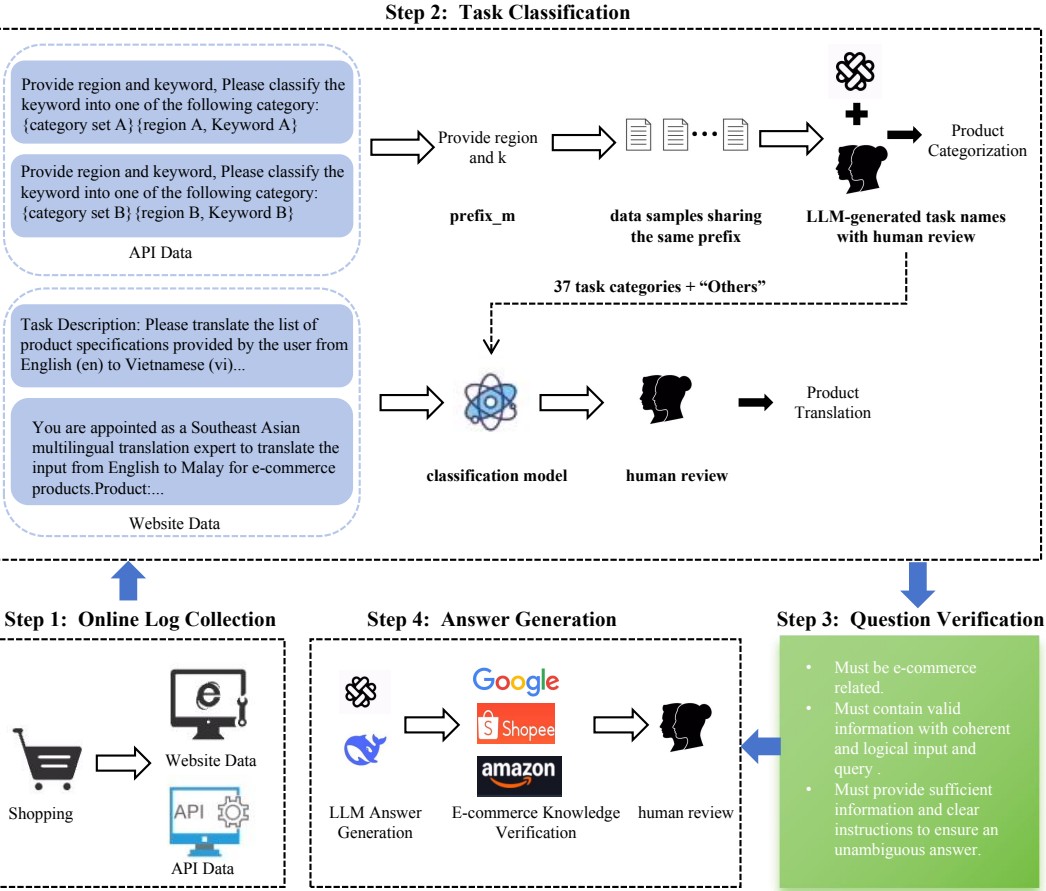

Figure 2: Overall pipeline of EcomEval dataset construction. The process consists of four stages: (1) collecting API call logs and website queries from LLM usage; (2) clustering API data via prefix grouping and classifying website data with a fine-tuned model to form 37 representative task categories; (3) verifying that sampled questions are e-commerce–relevant, coherent, and unambiguous; and (4) generating and fact-checking answers with LLMs, external sources, and human expert review across multiple languages.

ity to unknown or emerging user intents, a general "Others" category is introduced to capture novel or marginal requests not covered by the current classification label.

For model training, we select Qwen3-8B-Instruct as the base model, which demonstrates strong semantic understanding and generalization capabilities in e-commerce tasks and multilingual comprehension scenarios. To adapt the model to our classification task, we employ LoRA for parameter fine-tuning.

We also apply our classification model to categorize the website data, then sample a few data points from each category and manually annotate these samples to further verify the classification accuracy. Finally, we construct our evaluation set by selecting some samples from each category that have been manually confirmed.

**Step 3: Question Verification.** After collecting data from various categories using the aforementioned methods, we need to further review the quality of the questions. The questions must be e-commerce related and cover scenarios across the pre-sales and after-sales stages. Each question should contain sufficient and meaningful information. Assuming a question consists of input data and an instructional component, the semantics of both components must be coherent, logically consistent, and free of contradictions. Additionally, the information provided and the instructions given

in the question must be clear and comprehensive enough for an unambiguous answer. If the sampled data does not meet any of the criteria, it shall be deleted.

**Step 4: Answer Generation.** For the high-quality questions collected, we use LLMs to generate corresponding answers. For questions requiring specialized knowledge in e-commerce, such as product usage, brand information, or industry-specific content, we further verify the accuracy of the information using popular search engines such as Google, Bing, Baidu, as well as leading e-commerce platforms, including Shopee, Amazon, Lazada, and others. This may involve consulting existing blogs, official e-commerce stores, product pages, and user reviews to ensure factual correctness. Finally, human reviewers examine whether the generated answers follow all the instructions provided in the question and whether the answers are accurate, complete, and free from ambiguity.

In addition, we employ a team of over 50 human experts proficient in the target languages to manually review the quality of questions and answers in step 3 and step 4. These experts come from different countries and know the local culture and language well. They hold diverse educational backgrounds and have worked in the e-commerce field for many years.

## 4 EXPERIMENTS

### 4.1 EVALUATION METHOD

We have evaluated closed-source and open-source LLMs and Multimodal LLMs (MLLMs) on EcomEval tasks. We adopt LLM-as-a-judge, i.e., GPT-4.1, to evaluate the answers from each LLM/MLLM. Under our evaluation criteria, GPT-4.1 achieves over 90% agreement with human evaluators. We have tried using the open-source models Qwen2.5-14B and GPT-OSS-20B as evaluation models, both can serve as effective substitutes for GPT-4.1. LLM judge is asked to compare the model response to the reference answer in relation to the question and assign a score. We attach the 0-3 point scoring rubric in Appendix C. The score is reviewed by expert annotators to ensure the correctness of the evaluation. We adopt hierarchical averaging when reporting the performance of the models. We first compute the average score of all tasks within a category, and then the final score is obtained by averaging the category-level scores.

### 4.2 EXPERIMENT SETUP

We evaluate various closed-source and open-source LLMs and MLLMs, including GPT (OpenAI, Hurst et al. (2024)), Gemini (DeepMind), Qwen 2.5 (Qwen et al., 2025), Qwen 3 (Yang et al., 2025), and Llama (Meta, 2025) series. Note that for all open-source models, the instruction finetuned versions are used in our experiments. To maximize the reproducibility of the results, we set temperature as 0 for all models evaluated, including models from Qwen series, Llama series, GPT series, and Gemini 2.5 series. Since the evaluated models include both reasoning ("think") and non-reasoning ("non-think") architectures, to ensure a fair assessment—we incorporate few-shot prompts directly into the questions in the complex tasks; examples are provided in the Appendix B.2. As mentioned in Section 4.1, each model response is evaluated by the LLM judge on a 0-3 point scoring rubric and then reviewed by expert annotators. For analysis purposes, the 0-3 score is converted linearly to a percentage.

### 4.3 ANALYSIS AND CASE STUDY

#### 4.3.1 OVERALL PERFORMANCE

Overall, proprietary models, i.e., GPT models and Gemini models, perform better than open-source models. Specifically, GPT-5 has the best overall performance with an average score of 72.05%, GPT-4.1 comes close at 71.90%. GPT-5 tops the charts for tasks in the categories of Product Understanding and Shopping Reasoning, whereas GPT-4.1 is the best model for Ecom Services and Ecom Generation. The results show that Qwen3-32B outperforms other open-source LLMs and GPT-4o-mini. Although it is reported by Yang et al. (2025) that Qwen3-32B shows much better performance in general tasks than GPT-4o-mini, it is not the case in EcomEval, with Qwen3-32B (65.88%) performing just slightly better than GPT-4o-mini (65.19%). This shows that **the perfor-**

Table 2: Performance of Closed-Source and Open-Source LLMs on EcomEval Tasks. The best result in each category is in bold, and the second-best result is underlined.

| Models | Average | Ecom Generation | User Query & Review | Shopping Reasoning | Ecom Services | Product Understanding |
|---|---|---|---|---|---|---|
| **Closed-Source Large Language Models** | | | | | | |
| GPT-4o | 68.21 | 63.84 | 63.72 | 67.62 | 74.50 | 71.37 |
| GPT-4o-mini | 65.19 | 58.49 | 61.24 | 66.14 | 73.13 | 66.94 |
| GPT-4.1 | 71.90 | **69.14** | 66.03 | 72.70 | **75.15** | 76.49 |
| GPT-4.1-mini | 68.93 | 65.89 | 62.49 | 70.68 | 73.72 | 71.84 |
| GPT-5 | **72.05** | 68.71 | 66.07 | **74.82** | 73.53 | **77.10** |
| Gemini-2.5-pro | 69.99 | 66.61 | **66.83** | 68.41 | 74.09 | 74.00 |
| Gemini-2.5-flash | 68.87 | 66.06 | 65.89 | 66.74 | 73.02 | 72.64 |
| Average | | 65.54 | 64.61 | 69.59 | 73.88 | 72.91 |
| **Open-Source Large Language Models** | | | | | | |
| Qwen2.5-14B | 61.67 | 53.60 | 60.55 | 60.09 | 70.39 | 63.71 |
| Qwen2.5-72B | 65.31 | 58.69 | 61.99 | 64.54 | 73.13 | 68.19 |
| Qwen3-14B | 64.30 | 55.06 | 61.82 | 62.70 | 71.91 | 70.00 |
| Qwen3-32B | 65.88 | 58.80 | 63.19 | 65.86 | 71.32 | 70.20 |
| Qwen3-30B-A3B | 62.09 | 53.06 | 58.86 | 63.30 | 69.99 | 65.24 |
| LLaMa3-8B | 51.09 | 29.87 | 43.04 | 55.15 | 63.48 | 63.91 |
| LLaMa3-70B | 61.98 | 52.54 | 59.28 | 62.70 | 70.20 | 65.16 |
| LLaMa4-scout | 61.88 | 54.45 | 61.39 | 59.87 | 70.07 | 63.61 |
| Average | | 52.01 | 58.77 | 61.78 | 70.06 | 66.25 |

**mance in general benchmarks does not generalize to domain-specific tasks**, and **our benchmark can guide the development of LLMs in the e-commerce domain.**

### 4.3.2 MULTILINGUAL PERFORMANCE

The average score by language is presented in Table 3. While the gap between the best model and the second-best model is small for tasks in English (0.8%), the difference becomes significant in languages such as Indonesian (3.37%) and Malay (2.28%). Qwen3-32B performs notably better in Chinese, leading Llama4-scout by 6.65%. However, Llama4-scout achieves better results in Malay with 4.06% higher than Qwen3-32B. The variance of the performance shows the importance of having multilingual questions in LLM benchmarks to evaluate the multilingual capabilities of the models. **Overall, all models perform better for tasks in English as compared to tasks in low-resource languages.**

### 4.3.3 E-COMMERCE PERFORMANCE

In this section, we discuss the e-commerce performance of the models by analyzing the categories where the models do not perform well. From Table 2, we observe that both closed-source and open-source models perform poorly in e-commerce generative tasks. Particularly, the models underperform in product tag generation and product title generation tasks, **missing selling points in the product tag/title generation.** Another category where the models do not perform well is User Query&Review. **The models often overlook the information of product categories and targeted buyers when tested with tasks such as query-product matching and search relevance.**

To investigate the weakness of models in e-commerce tasks, we performed a failure analysis on incorrect cases under Ecom Generation. We randomly sample 100 failure cases and manually examine their underlying errors. The analysis revealed that most model failures fall into four main categories: (i) product content rule violations, (ii) product title length issues, (iii) product title formatting errors,

Table 3: Multilingual performance of Closed-Source and Open-Source LLMs on EcomEval Tasks. The best result for each language is in bold, and the second-best result is underlined.

| Models | English | Indonesian | Malay | Portuguese | Thai | Chinese | Vietnamese | Spanish |
|---|---|---|---|---|---|---|---|---|
| **Closed-Source Large Language Models** | | | | | | | | |
| GPT-4o | 72.73 | 62.89 | 64.10 | 67.29 | 67.34 | 71.85 | 65.82 | 71.95 |
| GPT-4o-mini | 69.93 | 59.26 | 63.61 | 66.93 | 62.64 | 66.93 | 64.46 | 67.10 |
| GPT-4.1 | **75.20** | 69.82 | 67.19 | 70.86 | **70.48** | **75.74** | 70.18 | 73.40 |
| GPT-4.1-mini | 72.47 | 67.13 | 65.07 | 67.76 | 67.20 | 71.33 | 66.64 | 72.80 |
| GPT-5 | 74.40 | **73.19** | **72.23** | **71.81** | 69.48 | 74.18 | **72.08** | **74.88** |
| Gemini-2.5-pro | 73.40 | 66.96 | 69.95 | 69.79 | 69.91 | 72.97 | 69.90 | 74.58 |
| Gemini-2.5-flash | 72.93 | 66.70 | 67.51 | 67.17 | 67.34 | 71.77 | 68.95 | 72.90 |
| **Open-Source Large Language Models** | | | | | | | | |
| Qwen2.5-14B | 66.47 | 55.88 | 59.22 | 62.40 | 59.65 | 66.93 | 59.16 | 55.60 |
| Qwen2.5-72B | 69.47 | 59.34 | 61.82 | 64.43 | 63.93 | 69.69 | 65.82 | 68.31 |
| Qwen3-14B | 69.73 | 57.18 | 61.66 | 63.83 | 64.21 | 70.12 | 62.83 | 65.00 |
| Qwen3-32B | 71.47 | 60.99 | 60.20 | 65.02 | 63.07 | 70.21 | 66.10 | 68.20 |
| Qwen3-30B-A3B | 68.00 | 57.52 | 58.73 | 60.74 | 60.65 | 66.67 | 60.38 | 64.30 |
| LLaMa3-8B | 62.80 | 52.76 | 48.16 | 52.64 | 53.95 | 54.75 | 47.05 | 48.10 |
| LLaMa3-70B | 68.40 | 55.36 | 61.98 | 63.60 | 60.08 | 63.82 | 60.65 | 62.20 |
| LLaMa4-scout | 68.80 | 54.06 | 64.26 | 62.29 | 63.50 | 63.56 | 60.52 | 64.50 |

and (iv) product title language mismatches. **Product Content Rules.** The model response does not fulfill user requirements, such as adding irrelevant or missing core product features (see Table E7). **Product Title Length.** The generated product title does not adhere to the length limit given by the user (see Table E8). **Product Title Formatting.** Format of the product title does not align with the rules given in the instructions (see Table E9). **Product Title Language.** The language used in the product title does not match the required language (see Table E10). The distribution of the error types is illustrated in Table 3: product content rules (52.4%), product title language (28.6%), product title length (14.3%), and product title formatting (4.7%).

Moreover, we observe that in Ecom Services, which includes the tasks of shopping guide and after-sales service (absent in existing e-commerce benchmarks), it is challenging for LLMs to tackle these questions from real-world online shopping scenarios. With LLMs scoring between 63%-76%, there is clear room for improvement in these tasks. These two tasks are representative in e-commerce applications because they reflect the ability of LLMs to guide buyers through pre-sales and post-sales stages. **The relatively modest performance in these areas highlights both the incompetence and the importance of strengthening LLM in shopping guide and after-sales tasks.** We include the bad cases of Ecom Services tasks in Tables E3, E4, E5, and E6.

### 4.3.4 MULTIMODAL PERFORMANCE

From Table 4, we can observe that GPT-5 and GPT-4.1 are the top two models in terms of average scores across all multimodal tasks. GPT-5 has the best performance in 4 tasks, namely product cover image selection (PCISel), product image content analysis (PICA), multimodal pickup reschedule detection (MPRD), and product comment & image summarization (PCISum). Gemini-2.5-flash outperforms other models in the tasks of multimodal product similarity (MPS) and multimodal search relevance (MSR), while GPT-4.1 comes out top for multimodal violation content detection (MVCD). Unexpectedly, LLaMa4-scout attains the highest score for multimodal brand recognition (MBR), surpassing all other models. We highlight that even proprietary models show weak results for the task MPS. When analyzing the results from this category, we find that the models tend to overestimate the similarity of two products by comparing their product titles, **overlooking subtle differences in their images**. This underscores **the importance of enhancing the multimodal capability of MLLM-based shopping assistants to achieve fine-grained product understanding**.

Table 4: Performance of Closed-Source and Open-Source MLLMs on EcomEval Multimodal Tasks. The best result of each task is in bold, and the second-best result is underlined.

| Models | Average | PCISel | PICA | MBR | MPS | MPRD | PCISum | MSR | MVCD |
|---|---|---|---|---|---|---|---|---|---|
| Closed-Source Multimodal Large Language Models | | | | | | | | | |
| GPT-4o | 62.44 | 71.33 | 45.89 | 70.33 | 55.01 | 63.11 | 61.67 | 68.16 | 64.03 |
| GPT-4.1 | 76.66 | 78.82 | 81.33 | 73.67 | 67.38 | 80.44 | 86.33 | 70.80 | **74.50** |
| GPT-5 | **77.04** | **81.20** | **83.09** | 77.67 | 62.23 | **82.00** | **87.33** | 71.60 | 71.20 |
| Gemini-2.5-pro | 74.44 | 78.48 | 79.58 | 78.00 | 63.95 | 66.67 | 86.00 | 70.54 | 72.30 |
| Gemini-2.5-flash | 73.27 | 78.82 | 80.28 | 71.67 | 68.07 | 59.78 | 82.67 | 75.04 | 69.82 |
| Average | | 77.73 | 74.03 | 74.27 | 63.33 | 70.40 | 80.80 | 71.23 | 70.37 |
| Open-Source Multimodal Large Language Models | | | | | | | | | |
| Qwen2-VL-7B | 47.29 | 67.93 | 57.82 | 47.67 | 60.16 | 43.33 | 21.67 | 31.92 | 47.78 |
| Qwen2.5-VL-7B | 48.05 | 71.33 | 70.46 | 31.00 | 47.79 | 42.67 | 25.33 | 42.23 | 53.56 |
| Qwen2.5-VL-72B | 60.82 | 71.67 | 79.58 | 57.33 | 63.95 | 56.89 | 37.00 | 53.34 | 66.79 |
| LLaMa3.2-11B-Vision | 37.20 | 43.78 | 59.93 | 28.67 | 31.64 | 25.78 | 25.33 | 38.53 | 43.92 |
| LLaMa4-scout | 59.92 | 75.76 | 70.11 | **81.67** | 47.11 | 43.33 | 51.33 | 56.25 | 53.84 |
| Average | | 66.09 | 67.58 | 49.27 | 50.13 | 42.40 | 32.13 | 44.46 | 53.18 |

## 5 CONCLUSION

This paper presents EcomEval, a multilingual and multimodal e-commerce benchmark covering 6 primary categories and 37 tasks across 8 languages, providing comprehensive real-world use cases in the e-commerce domain. From the average scores of each question and category, we derive difficulty tags at the question-level and category-level, respectively. The difficulty tags are included in our open-source benchmark. Our benchmark data mainly comes from real internal e-commerce scenarios, and releasing it publicly requires management approval. Currently, only 7,200 questions have been approved for open sourcing. More data will be released gradually, with an expected total exceeding 20,000 questions.

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

# A    ECOMEVAL DATA COMPOSITION

We show a detailed count of the number of questions for each task in Table A1.

Table A1: Distribution of EcomEval by tasks.

| Category | Task | Number of Samples |
|---|---|---|
| Product Understanding | Product Inquiry | 358 |
| | Product Categorization | 126 |
| | Product Type Understanding | 256 |
| | Product Attribute Extraction | 375 |
| | Prohibited Item Identification | 100 |
| | Brand Knowledge | 101 |
| | Brand Model | 248 |
| Ecom Services | Shopping Guide | 187 |
| | After-Sales Service | 397 |
| | Order Classification | 118 |
| | Logistic Dialogue Classification | 108 |
| | Fraud Risk Analysis | 137 |
| User Query & Review | Query Understanding | 246 |
| | Query-Product Matching | 354 |
| | User Review | 361 |
| | Review Sentiment Classification | 99 |
| | Search Relevance | 119 |
| Shopping Reasoning | Product Recommendation | 329 |
| | Numerical Reasoning | 270 |
| | Similar Product Identification | 329 |
| | Description Similarity Evaluation | 355 |
| | Complementary Product Recommendation | 131 |
| | Selling Point Analysis | 99 |
| Ecom Generation | Product Information Generation | 210 |
| | Product Title Generation | 103 |
| | Product Title Refinement | 206 |
| | Product Tag Generation | 131 |
| | After-Sales Issue Summarization | 100 |
| | Product Translation | 358 |
| Ecom Multimodal | Product Cover Image Selection | 100 |
| | Product Image Context Analysis | 95 |
| | Multimodal Brand Recognition | 100 |
| | Multimodal Product Similarity | 97 |
| | Multimodal Pickup Reschedule Detection | 150 |
| | Product Comment & Image Summarization | 100 |
| | Multimodal Search Relevance | 126 |
| | Multimodal Violation Content Detection | 121 |

Details of the distribution of question counts across the eight languages can be found in Table A2.

Table A2: Distribution of EcomEval by languages.

| Language | Number of Samples |
|---|---|
| English | 2051 |
| Indonesian | 802 |
| Malay | 587 |
| Portuguese | 818 |
| Thai | 594 |
| Chinese | 1143 |
| Vietnamese | 605 |
| Spanish | 600 |

## B EXAMPLES OF ECOMEVAL'S TASKS

### B.1 PRODUCT UNDERSTANDING - BRAND KNOWLEDGE

#### B.1.1 QUESTION

你是一个品牌合规检测专家，按以下规则处理输入数据
你是一个擅长品牌识别和文本分析的助手。请提取出{title}中产品符合{category_path}特征的自身品牌。
输入内容:[{"category_path": "商品绑定的完整类目路径范围依次缩小","brand":"品牌名" "title": "商品的标题信息"}]
处理规则：
规则一：
步骤1) 在{title}中寻找可能的品牌信息
步骤2) 如果提取到关键词是{brand}的简称、缩写、错误写法、子品牌、母公司、合作伙伴、竞争对手时输出N,没有则输出Y
规则二：只使用{category_path}从{title}中提取品牌
步骤1) 逐步分析{category_path}中每个类目的特征
步骤2) 在{title}中寻找主营产品符合该类目特征的品牌
步骤3) 排除验证：当候选词同时满足以下条件时保留
• 非产品型号特征
• 非营销术语（排除旗舰版/专业版等）
• 候选品牌与当前产品不存在配件/耗材类关系
步骤4) 这里禁止判断与{brand}的关系，单独判断候选词是否是明确的品牌，如果是独立品牌则输出N。未找到明确品牌信息则输出Y
注意：
- 必须严格按照规则中每个步骤进行处理，
- 分别处理规则一和规则二并保存处理结果
- 只有规则一和规则二的判断结果都是Y时，res输出Y，其余情况输出N
- 若提取到到品牌信息不确定是否是独立品牌则在brand中额外输出+可能是品牌+
输出要求：
- 用JSON数组格式严格输出[{"res":"Y/N","brand":"仅N时存在,标题中提取到的品牌信息","explain":"每一步完整的推理过程"}]
- explain需用中文详细说明分析步骤
- 禁止输出除json外额外内容
现在处理这个输入：
{"title": "Used shuttlecock Gred A, B & C", "category_path": ["Sports & Outdoors", "Sports & Outdoor Recreation Equipments", "Badminton", "Shuttlecocks"], "brand": "Felet"}

#### B.1.2 REFERENCE ANSWER

[{"res": "Y", "explain": "规则一分析：标题'Used shuttlecock Gred A, B & C'中未提取到品牌信息Felet，规则一输出Y。规则二分析：各类目在标题中没有提取到任何品牌信息，规则二输出Y。最终两规则均Y故res=Y"}]

## B.2 ECOM SERVICE-LOGISTIC DIALOGUE CLASSIFICATION

Here is an example of logistic dialogue classification.

### B.2.1 QUESTION

You are a helpful assistant that's experienced with customer service of an online e-commerce plat-form. You provide factual, accurate, professional and valuable responses.Given a conversation be-tween a user and the customer service live agent, determine which of the listed categories is related to the user's question, and response with its ID. (Each category is in presented in the format of "[ID] L1 > L2 > L3", where "ID" is the category ID and "L1 > L2 > L3" is a 3-layer structure, with in-creasing granularity. The user question's is said to be related to a category if and only if it is related to all L1, L2 and L3.) If none of the category is related to the user's question, then return "NA".

===== Example 1 =====

[Conversation]

[AGENT] Hi gymsportzfitness, thanks for contacting us. Our customer service representative will assist you shortly.

[AGENT] Welcome! This is Janzel, and I will be assisting you today.

[USER] Hi, orders are not collected yesterday

[USER] Please arrange for driver to pickup the orders by today

[AGENT] I understand that you are concerned about the pickup for your orders. No worries; I will try my best to assist you with this.

[AGENT] May I have 1 order ID, pleasE?

[USER] 1 of order is 240621KKHVBJTP

[AGENT] Thank you for providing the order ID.

[AGENT] Let me check on this real quick. Do you mind if I put you on hold for 2 minutes?

[USER] sure

[AGENT] Thank you so much for the wait.

[AGENT] I know it is important for you to have this issue resolved. I will inform the team to come pick up your item as soon as possible.

[AGENT] No worries; I will be handling this case with high priority, and your item will be picked up. You can count on me to help you with your problem as one of our valued customers.

[AGENT] I will follow up with our fleet team to arrange a pickup. For now, please anticipate the pickup of the parcel within 24 to 48 business hours for a timeframe [USER] Thank you.

[AGENT] You're most welcome, and I'm happy to know that we were able to assist you.

[AGENT] Is there anything else I can assist you with? It's my pleasure to be of assistance.

[Categories]

- [1423] Logistics > To ship > Rider did not show up/late for pickup

- [8471] Return/Refund > Raising disputes for return/refund requests (NEW flow for selected sellers) > none

- [1849] Seller Operations > Seller Centre / My Shop App > Why is my product deleted/suspended?

[Answer] 1423

===== End of Example 1 =====

===== Example 2 =====

[Conversation]

[AGENT] Hi aromaserapi, thanks for contacting us. Our customer service representative will assist you shortly.

[AGENT] My name is Char, and I will be assisting you today.

[AGENT] Hi, aromaserapi!

[USER] I would like to enquire about the status of my refund

[AGENT] I understand your concern that you want to inquire about your refund status; please let me help you with this.

[AGENT] May I have the order ID of the issue?

[USER] 1799234087724548181

[AGENT] May I confirm that this is the order ID of the issue: 24060466CC4J6K?

[USER] Yes

[AGENT] Please allow me to put you on hold for 2 minutes while I do some quick checking on this matter.

[USER] Okay

[AGENT] Thank you for patiently waiting. Upon checking here. Due to the Lost Parcel by SPX Xpress, you [will be receiving reimbursement for the lost parcel in your Seller Balance [within 7-14 working days.

[USER] Yes. Its more than 14 working days?

[AGENT] 7 to 14 working days. Yes, but not to worry; it is only a timeframe, and we will come back to you as soon as possible.

[USER] When will it be refunded?

[USER] Today?

[AGENT] Yes.

[USER] Okay.

[USER] Yes, continue

[USER] It will refunded by end of today?

[AGENT] That is correct. In the event that you do not receive the refund within this timeframe, you can contact us again for your concern.

[USER] Okay. I will check

[AGENT] Sure! Is there anything else I can assist you with? It's my pleasure to be of assistance.

[Categories]

- [8617] Logistics > Delivered > Shipping fee related claim

- [1657] General > Seller Penalty Points System > Penalty appeals

[Answer] NA

===== End of Example 2 =====

Now, think carefully and reply with the related category ID. (Do not provide any additional information.)

[Conversation]

[AGENT] Good day, christianherrera09!Kumusta? I'm Jay Marie. How can I help?

[AGENT] I understand you want to expedite the delivery for this order. Did I understand correctly?

[USER] I think the delivery driver stole my package

[AGENT] I understand that you have concerns regarding sa inyong order that it seems like the rider stole it. No worries! I'd be more than happy to check on this for you.

[USER] She marked it as delivered but didn't give the parcel

[AGENT] May I know the order ID?

[USER] I have another parcel coming and i think she's the driver again

[AGENT] Thank you for the order number. Check ko lang po muna ito. Can I put this chat on hold for 1-2 minutes , then I'll be right back with the update?

[USER] I hope this doesn't happen again

[USER] Sure

[AGENT] Thank you for waiting po! As per checking po on my end, your parcel has been delivered, pero walang order na dumating. Just for confirmation po, nakita nyo po ba ang proof of delivery at familiar ba kayo sa location ng picture?

[USER] There's no proof of delivery and that's so suspicious. They must always provide one

[USER] Wala pong pic

[AGENT] I see, thank you for letting me know.

[AGENT] May I verify the address if tama po?364, Sapang I, Ternate, South Luzon, Cavite, 4111

[USER] Delivered lang po tlga kaya kinabahan po ako

[USER] Yes po

[USER] Usually dumadating naman po kase ilang beses na kong nag order sa with the same address

[USER] May contact information din po ako so dapat nag call nalang sya kung d po mahanap

[AGENT] Salamat po sa lahat ng information.

[AGENT] In this case po, we have two options: either ma contact niyo po si rider kung saan niya nailagay or naibigay ang order niyo, baka honest mistake lang ito, or i-dispute po natin ang proof of delivery at gawan ito ng escalation report para ma imbestigahan.

[USER] Report po sana

[USER] Wala kase syang contact information kaya d ko po matanong sakanya directly

[AGENT] If you wish po, na malaman na rin po ang contact number po ng rider na may hawak po ng ating parcel para ma-coordinate ninyo? No worries, I will get it for you po. Just let me know lang po if gusto niyo matawagan.

[USER] Yun po tlga sana gagawin ko since hassle po yung ganito

[USER] Kung pwede po sana

[AGENT] Here's the number po ni rider: 09217024560 para madirect contact niyo po siya.

[AGENT] It's important po to note that our riders might be occupied with driving or fulfilling other orders, which may make it difficult po for them to answer your message or phone call.

[AGENT] It's essential po to adhere to the regulations outlined in the Data Privacy Act of 2012 and refrain from any misuse of this information po.

[AGENT] Feel free to contact po si rider and if ever unresponsive po si rider, just feel free to contact us back po.

[USER] My problem is solved, end chat

[Categories]

- [2015] Logistics > Delivered > Status is delivered but not yet received

- [3045] Logistics > Order Cancellation > Order is shipped but cancelled in system

- [2661] Return/Refund > Buyer shipped return parcel > How long will it take for me to receive my refund?

- [3111] Logistics > To receive > Delivered but status is not updated

- [2656] Logistics > Order Cancellation > Order cancelled by seller

- [3179] Only for TFE use > Don't put into DS model training > Only for TFE use - Intent 2023

[Answer]

### B.2.2   REFERENCE ANSWER

2015

## B.3   USER QUERY & REVIEW - REVIEW SENTIMENT CLASSIFICATION

### B.3.1   QUESTION

Please classify the delivery experience and sentiment in the customer's comment below.
- Delivery Experience: Describe rider behavior, delivery speed, overall delivery quality, etc.
- Sentiment: Choose only one: Positive, Neutral, or Negative.
Customer comment: "User Tips:Buy it now, ypu won't regret it the product is good
Packaging:The product is safe coz' the item is packaged is well.
Beauty Profile:Perfect for my skin tone
Received item safe coz'it's packed well. Thank you seller and also to rider."
Respond in this format:
Delivery Experience: <short description>
Sentiment: <Positive/Neutral/Negative>

### B.3.2   REFERENCE ANSWER

Delivery Experience: The delivery experience was positive, with the rider being acknowledged and the item being received safely due to good packaging.
Sentiment: Positive

## B.4   SHOPPING REASONING - PRODUCT RECOMMENDATION

### B.4.1   QUESTION

I provide a product description '1. Butt Lifter Slimming High Waist Girdle Corset Long Shaper Girdle Pants Plus Size Girdle Shapewear BengkungColour Black Nude Size M L XL 2XL 3XL 4XL Measurement Size M L Waist 65 80 CM 26 32 inch Weight 50 65 KG Size XL XXL Waist 80 100 CM 32 40 inch Weight 65 80 KG Size 3XL Waist 100 120 CM 40 47 inch Weight 80 100 KG Size 4XL Waist 120 135 CM 47 53 inch Weight 100 115 KG Material Spandex and Nylon Slim your Tummy Waist and Bottom and wear figure hugging clothes with new confidence Breathable and Comfortable material Wear All day along 2. Pants Girdle Plus Size Corset Girdle Slimming Girdle Shapewear Borong Bengkung High Waist Girdle t Welcome to my shop nMalaysia local seller nFollow the store to get more real time data of store products nWe ship daily order today and ship immediately n100 brand new high quality product n nFeatures nColor Black Beige nSize M L XL XXL 3XL 4XL n1 on waist spiral steel bone to prevent curling n2 breathable comfortable stretchable healthy no smell perfect design n3 butt lifter with tummy control waist trainer waist slimming n4 high waist butt lift panties n5 corrective slim underwear n6 Suitable season four seasons nSize of the prodcut Same as the picture shown nSize Waist Weight nM L 20 28 40 50KG nXL 2XL 26 31 50 60KG n3XL 30 35 60 70KG n4XL 33 40 70 80KG nPackage Includes 1pcs Women Control Panties i cn 11134208 7ras8 m4r51le1xjrd69 1 000 i cn 11134208 7ras8 m4r51le1yybt0d 1 000 i cn 11134208 7ras8 m4r51le20cw9b1 1 000 i cn 11134 3. ANGELMOMMY Double-Breasted High-Waist Abdomen Tucking Underwear Body-Shaping Belly-Shaping Pants Women Hip Lifting 039 t Welcome to angelmommy os where we strive to provide our customers an array of products that satisfy demand for quality and efficiency Buy our products in a worry free manner as we guarantee

100 authenticity Shopping can t get any easier than this so start today n n Product features n Six advantages n Focus on abdominal waist n Improve abdominal relaxation n Correction of pelvic expansion n Shape sexy curves n Three dimensional hip lift n Shape legs n n Product effect n Flat Belly flat belly n Waist sculpting tighten fat n Butt lift curve butt lift n Slim legs shape legs n n Product details n Three breasted buckle for more shape n Upgraded three breasted belly pants n Through the continuous pressurization of the buckle n Get rid of belly fat n Make your belly thinner and thinner n Bring you back to your graceful posture n n Product si' for you. Please analyze the product names mentioned and recommend up to 10 complementary items to enhance the user's purchase experience. Focus on relevant and useful complementary items that naturally pair with the main product. Ensure the recommendations follow these rules:

1. Exclude included accessories.
2. Ensure compatibility (brand, model, category).
3. Avoid items of the same type as 'Shapewear'.
4. Exclude shipping/packaging items.
5. Rank by purchase co-occurrence.
6. Avoid sets containing 'gurdle'.

Generate keyword list for the recommended items, using 'Malaysia Malay' language whenever possible:

'keywords': Various related keywords for practical items.

Ensure diversity and relevance of the keyword while avoiding duplication. Output in JSON format: "keywords": [...], do not return any other texts

### B.4.2 REFERENCE ANSWER

"keywords": ["baju dalam wanita", "bra tanpa dawai", "bralette selesa", "pakaian dalam seamless", "stokin wanita", "pakaian tidur wanita", "losyen pelangsing badan", "pencuci pakaian dalam", "pakaian senaman wanita", "minyak urut pelangsing"]

## B.5 ECOM GENERATION - PRODUCT TITLE TRANSLATION

### B.5.1 QUESTION

Task Description: Please translate the list of product specifications provided by the user from English (en) to Vietnamese (vi). Input Format: 1. Multiple specifications enclosed in square brackets (e.g., [Specification 1, Specification 2, Specification 3]), with each specification separated by a semicolon (;).

Translation Requirements: 1. Format Preservation: The output format should be similar to the input format, maintaining the original list structure, with semicolons used as separators. 2. Output Content: Only output the translated specifications, without explanations or any additional content. 3. Accuracy: Ensure the translations are as accurate as possible.

[Anti-pilling: Green/White+Pink/White+Blue/White; Anti-pilling: Green/White+Pink/White+White/Yellow; Anti-pilling: White/Card+White/Blue+White/Yellow; Anti-pilling: White/Blue+White/Yellow+Pink/White; Anti-pilling: Green/White+White/Blue+White/Card; Anti-pilling: Green/White+Blue/White+White/Yellow+White/Card; Anti-pilling: Blue/White+White/Yellow+White/Card+Pink/White; Anti-pilling: Green/White+Blue/White+White/Blue+White/Yellow; Anti-pilling: Green/White+Green/White+White/Blue+White/Blue; Anti-pilling: Green/White+Green/White+White/Yellow+White/Yellow; Anti-pilling: White/Yellow+White/Yellow+Pink/White+Pink/White; 6 styles, 1 pair each; 3 pairs, random if no note; 4 pairs, random if no note]

### B.5.2 REFERENCE ANSWER

[Chống xù: Xanh/Trắng+Hồng/Trắng+Xanh/Trắng; Chống xù: Xanh/Trắng+Hồng/Trắng+Trắng/Vàng; Chống xù: Trắng/Thẻ+Trắng/Xanh+Trắng/Vàng; Chống xù: Trắng/Xanh+Trắng/Vàng+Hồng/Trắng; Chống xù: Xanh/Trắng+Trắng/Xanh+Trắng/Thẻ; Chống xù: Xanh/Trắng+Xanh/Trắng+Trắng/Vàng+Trắng/Thẻ; Chống xù: Xanh/Trắng+Trắng/Vàng+Trắng/Thẻ+Hồng/Trắng; Chống

xù: Xanh/Trắng+Xanh/Trắng+Trắng/Xanh+Trắng/Vàng; Chống
xù: Xanh/Trắng+Xanh/Trắng+Trắng/Xanh+Xanh/Trắng; Chống
xù: Xanh/Trắng+Xanh/Trắng+Trắng/Vàng+Vàng/Trắng; Chống xù:
Trắng/Vàng+Vàng/Trắng+Hồng/Trắng+Hồng/Trắng; 6 mẫu, mỗi mẫu 1 đôi; 3 đôi, ngẫu nhiên nếu
không ghi chú; 4 đôi, ngẫu nhiên nếu không ghi chú]

## B.6 Ecom Multimodal

### B.6.1 Question

Analyze the image and identify the brand of the product. If the brand matches one from the list below, return
only the exact brand name(s). If no match is found or the brand is not in the list, return "-".

Accepted brands: Calvin Klein, Chanel, John Deere, Levi's, Nike, Tommy Hilfiger, Zimmermann, Lacoste,
Oakley, Osklen, Burberry, Estee Lauder, Giorgio Armani, Gucci, Hermés Paris, Hugo Boss, Lancôme, Louis
Vuitton, New Balance, Prada, Puma, Ralph Lauren, Ralph Lauren Fragrances, Tiffany & Co., Valentino, Ver-
sace, Yves Saint Laurent, Abercrombie, Adidas, Casio, Coach, Converse, Denizen, Diesel, Dockers, JanSport,
Jimmy Choo, Jordan, Kiehl's Since 1851, Kipling, Maison Margiela Fragrances, Marc Jacobs, Maybelline
New York, Michael Kors, New Era, Nioxin, Supreme, The North Face, Timberland, Under Armour, Vans,
Victoria's Secret, Viktor&Rolf, Ray-Ban, Vogue Eyewear, Persol, Oliver Peoples, Arnette, Costa Del Mar,
Emporio Armani, Dolce & Gabbana, Miu Miu, Ferragamo, Bvlgari, DKNY, Fendi, Mizuno, Asics, CeraVe,
Dior, Azzaro, Bozzano, David Beckham, Dickies, Gabriela Sabatini, Harley-Davidson, Joop!, Kate Spade, Mu-
gler, Redken, SkinCeuticals, Urban Decay, WaterPik, G-Shock, Vichy, Flamengo, Bayer, São Paulo, Palmeiras,
Corinthians, Santos, Atlético-MG, Athletico-PR, Fluminense, Botafogo, Fortaleza, Grêmio, Bahia, Interna-
cional, América-MG, Vasco, Manchester City, Real Madrid, Bayern Munich, Liverpool, Paris Saint-Germain,
Inter Milan, Chelsea, Barcelona, Manchester United, Juventus, Rolex, Olympikus, Crocs, Zara, Vick, Wahl,
Kenner, Loewe, Celine, Montblanc, Dyson, Balenciaga, Panerai, Van Cleef & Arpels, Bottega Veneta, Chloe,
Piaget, Cartier, IWC, Pandora, Carolina Herrera, Barbie, Stanley, Bobbie Goods, Fila, Minecraft, Havaianas,
PopMart/Labubu. Answer can only be the brand or "-", nothing else.

### B.6.2 Input Image

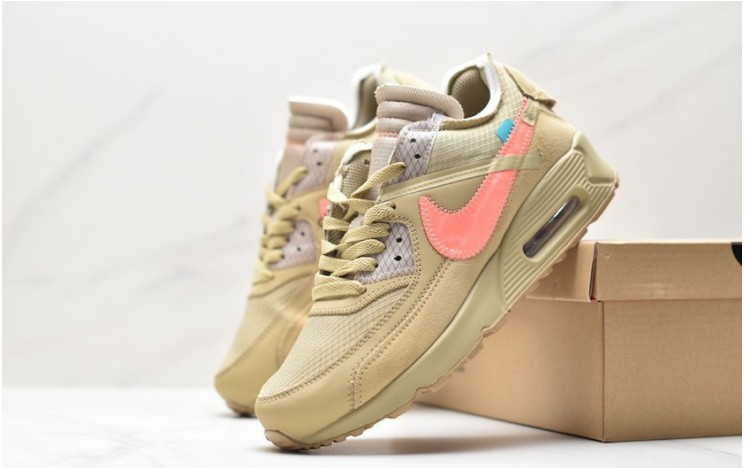

### B.6.3 Reference Answer

Nike

## B.7 Shopping Guide – multi-turn questions

Note that the assistant's answer may not correct.

**user**: I think the Li-Ning N90 III is pretty good. How much does it cost?

**assistant**: I'm sorry, but I'm unable to provide price information for the Li-Ning N90 III because my knowledge
base was last updated in April 2023, and I cannot access the latest market prices. I suggest checking sports
equipment stores or official authorized online platforms for the most accurate pricing information.

Table C1: Scoring Rubric for Model Responses

| Score | Description |
|---|---|
| 3 | The response is entirely correct. |
| 2 | The response is nearly correct but has flaws. |
| 1 | The response is incorrect. |
| 0 | The response is off-topic or violates safety guidelines. |

**user**: What about the Victor Jetspeed S10?

**assistant**: The Victor Jetspeed S10 is a badminton racket designed for offensive and control-oriented players, known for its lightweight design and high swing speed. However, I cannot provide specific price information, as it may vary by region, retailer, and time. I recommend checking the Victor official website or authorized Victor badminton racket dealers for the most accurate pricing information.

**user**: What if buying it in Beijing?

**assistant**: If you are in Beijing, I suggest visiting the official authorized Victor badminton racket store or sports equipment shops located at Beijing's Chaoyang District, Workers' Stadium area to purchase the Victor Jetspeed S10. These stores usually offer genuine products and professional after-sales service. You can also look for Victor authorized stores or dealers on online shopping platforms such as JD or Tmall to get more accurate pricing and purchasing information.

**user**: Is there anything cheaper, like in the 200–300 RMB range?

**assistant**: In the 200–300 RMB price range, you can consider some cost-effective badminton racket brands such as Felet, Mizuno, and Apacs. These brands typically offer more affordable prices while still providing reasonable performance and durability. I recommend checking sports equipment stores or authorized online platforms to find a racket that fits your needs and budget. It's best to try hitting with the racket before purchasing to ensure it suits your personal preferences and playing style.

**user**: I want to choose one lightweight and powerful racket from the three you recommended.

**assistant**: I'm very sorry, but I can't directly recommend a specific product. Everyone's needs and preferences differ, and the definition of "lightweight and powerful" may vary from person to person. When choosing a badminton racket, I suggest considering the following factors:

Weight: A lighter racket allows for more flexible swings, but make sure it still has enough power and stability.

Balance point: A head-heavy racket suits offensive players, offering more power and attacking capability; a head-light racket is better for defensive or control-oriented play.

Stiffness: A stiffer frame provides more power and control, while a more flexible frame is better for control and precision.

Grip size: Choose a grip size that matches your hand shape to ensure comfortable and stable handling.

I recommend going to a sports equipment store to try hitting with a few rackets in person and choose based on your personal needs and preferences.

## C  LLM JUDGE SCORING RUBRIC

The scoring rubric used in our experiments is presented in Table C1.

## D  TASKS ACROSS DIFFERENT DIFFICULTY LEVELS

The examples of tasks across different difficulty levels are shown in Table D1. We observe that easy tasks typically assess the capabilities of a model in a straightforward manner, where answers can be derived through simple reasoning from the given information. Most medium-difficulty tasks evaluate the model's grasp of e-commerce-specific knowledge, including product usage and features tailored to Southeast Asian markets, as well as regionally prevalent brand models. Beyond this foundational knowledge, we also assess the model's understanding of after-sales service — such as returns, exchanges, and logistics — testing not only its knowledge of the products themselves, but also its ability to reason through real-world post-purchase processes. Hard tasks not only evaluate the model's mastery of Southeast Asian e-commerce knowledge, but also demand its ability to follow complex instructions to accurately process inputs and generate appropriate outputs.

Table D1: Examples of tasks across different difficulty levels

| task | key challenges | average | difficulty level |
|---|---|---|---|
| order classification | Given an order name, output the category to which the order belongs. | 86.89 | easy |
| review sentiment classification | Given the user review, infer the user's sentiment type: negative, neutral, or positive. | 81.16 | easy |
| product categorization | Given the product name and the list of all product categories, output the correct category for the product. | 73.46 | medium |
| similar product identification | Determine the relationship between Product 1 and Product 2 and whether they are similar. | 73.73 | medium |
| product title refinement | Require the model to process input based on complex instructions, and generate a revised title in the specified format and language. | 54.57 | hard |
| search relevance assessment | This task requires assessing the search relevance scores of multiple products within a single query. The output format follows strict specifications, and the data classification logic across different relevance levels is complex, with detailed and precise rules. | 63.6 | hard |

Additionally, we assign a difficulty level to each individual question based on its average performance across all models. The detailed difficulty labels for each question can be found in the "difficulty level" column of our open-sourced dataset.

# E    BAD CASE ANALYSIS

## E.1    USER QUERY & REVIEW

We included the bad case in Table E1.

Table E1: Bad Case in User Query & Review Category

**Category: User Query & Review**

**Question**: A user would like to find a product with the keyword 'brass tray rectangular'. What is the relation between the user's intended products and the product boweiwj Stainless Steel Towel Tray Storage Tray Tray Dish Plate Tea Tray Fruit Trays Cosmetics Jewelry Organizer Gold Oval Tray 9in"?
A. substitute
B. irrelevant
C. exact match
D. complement

**Correct answer**: A

**Gemini-2.5 Pro Response:**
Let's break this down:

User's intended product:
* **Material:** Brass
* **Type:** Tray
* **Shape:** Rectangular

Product 'boweiwj Stainless Steel Towel Tray... Gold Oval Tray 9in':
* **Material:** Stainless Steel
* **Type:** Tray (multiple uses listed)
* **Shape:** Oval
* **Color:** Gold

Comparison:
1. **Material:** User wants "brass," product is "stainless steel." (Mismatch)
2. **Type:** User wants "tray," product is a "tray." (Match)
3. **Shape:** User wants "rectangular," product is "oval." (Mismatch)

4. **Color/Appearance:** Product is "gold" colored, which can sometimes be visually similar to brass, but "brass" is a specific material, not just a color.

Considering the options:
* **A. substitute:** A substitute should be able to perform the same function and be reasonably similar. While it's a tray, the material and shape are different. The gold color might make it a *visual* substitute for some, but "brass" specifies a material. It's a weak substitute at best due to material and shape differences.
* **B. irrelevant:** The product is a tray, which is what the user is looking for in general. However, it fails on two key specific attributes: material (brass vs. stainless steel) and shape (rectangular vs. oval). If the user *specifically* needs brass and rectangular, then this stainless steel oval tray is not relevant to those specific needs.
* **C. exact match:** Clearly not an exact match due to differences in material and shape.
* **D. complement:** This product is not something typically used *with* a brass rectangular tray; it's an alternative type of tray.

The product is a tray, so it's not *completely* unrelated. However, the user has specified "brass" (a material) and "rectangular" (a shape). The offered product is "stainless steel" (a different material) and "oval" (a different shape). Because two of the three explicit keywords describing the desired item are mismatched, the product is not a good fit.

While it is a tray, the deviation in *specified* material and shape makes it quite different from the user's request. If a user explicitly asks for "brass" and "rectangular," offering "stainless steel" and "oval" is more likely to be perceived as irrelevant to their specific need than a suitable substitute.

Therefore, the most appropriate answer is **B. irrelevant** because it fails to meet the key specific attributes requested by the user (material and shape), even though it matches the general product category (tray).

The final answer is $\boxed{B}$

**LLM Judge**:
The model response chooses "irrelevant" (B) rather than the correct answer "substitute" (A), despite a detailed comparison; thus, it is incorrect.

**Final Score**: 1

### E.2 ECOM GENERATIVE ABILITY

We included the bad case in Table E2.

Table E2: Bad Case in Ecom Generative Ability Category

**Category: Ecom Generative Ability**

**Question**: **Task:**
Integrate 'value' into Eucerin Skin Calming Cream / Skin Calming Lotion / Itch Relief Intensive Calming Lotion **without modifying origin content**. Output optimized title only.

—

**Processing Pipeline**

**1. Value Preprocessing**
- **Preservation Rules**:
- Retain ALL original symbols/emojis/formatting ( $<...>$, 3.5g, 20241014)
- Protect multilingual terms in their original form

- **Term Construction**:
- **SPF-exclusive combination**:
• **Only** combine 'key_info'+'value' **if** 'key_info="SPF"' (SPF+30→SPF30)

- **All other key_info**: Use raw 'value' directly without prefix
- Merge variants/ranges:
- Overlaps: '30,spf30+' → 'SPF30+'
- Ranges: '50ml+100ml' → '50/100ml'
- Deduplicate **per category** (SPF50+SPF50→SPF50)

- **Language Normalization**:
- Auto-translate to match adjacent terms' dominant language
- Preserve technical codes (PA++++/SPF50) across languages

**2. Insertion Validation**
- **Mandatory Skip Conditions**:
- Existing equivalent **in any language** (保湿=Moisturizing)
- Logical contradictions (Non-Oily vs Oily)
- Full value/key_info already present (skip if "SPF30" exists when inserting SPF30)

**3. Context-Aware Insertion**
- **Priority Positions**:
1. Adjacent to primary product type (e.g., _Sunscreen_ SPF50)
2. Between high-value term & benefit statement
3. Title end (last resort)

- **Linguistic Alignment**:
- Match grammar structure & language flow:
• Insert "保湿" not "Moisturizing" in Chinese-dominant segments
• Mirror adjective/noun order of surrounding text

- **Technical Formatting**:
- Cluster related specs (SPF50/PA++++ Protection)
- Maintain original casing except technical codes

**4. Absolute Constraints**
- NEVER:
• Remove/modify origin content
• Add conjunctions (with/for/by)
• Repeat existing specs in any language
• Insert 'key_info' text (except SPF technical codes)
• Break protected patterns (<Cosmetics>, 3.5g)

—

**Examples**
Valid:
• 'key_info:SPF + 30' → "Sunscreen SPF30 Cream"
• "Sunscreen SPF50/PA++++ 100ml" (clustered specs)
• 'key_info:SkinType + All skIN' → "Lotion All Skin Face Cream" (no "SkinType" prefix)

Invalid:
• "Moisturizing 保湿Cream" (cross-language duplicate)
• "For SPF50 Milk" (illegal preposition)

—

**Output Format**
Return ONLY the optimized title without explanations.

—

**Input**:
- title: Eucerin Skin Calming Cream / Skin Calming Lotion / Itch Relief Intensive Calming Lotion
- key_info: Body Care Benefits
- value: Tender care

**Reference answer**: Eucerin Skin Calming Cream / Skin Calming Lotion / Itch Relief Intensive Calming Lotion Tender Care

**GPT-4.1 Response**: Eucerin Skin Calming Cream Tender care / Skin Calming Lotion Tender care / Itch Relief Intensive Calming Lotion Tender care

**LLM Judge**: The model response incorrectly splits and inserts "Tender care" after each variant, altering the original grouped structure rather than adding the value at the title end, thus violating the constraint to not modify origin content.

**Final Score**: 1

### E.3    ECOM SERVICES

#### E.3.1    SHOPPING GUIDE

We included the bad case in Table E3

Table E3: Bad Case in Task Shopping Guide

| Category: Ecom Services - Shopping Guide |
| --- |
| **Question**: Which feature should be considered for comfort and fit in a formal suit?
A. Color and patterns
B. Trousers cut and break
C. Removing or adding padding to shoulders
D. Number of buttons and lapel type |
| **Correct answer**: C |
| **GPT-5 Response**: B. Trousers cut and break
These directly affect how the suit sits and moves on your body, impacting comfort and overall fit. |
| **LLM Judge**: The model response chooses B, which relates to fit but not comfort in the same direct way as shoulder padding (C), which is more critical; thus, it's nearly correct but not the best answer. |
| **Final Score**: 1 |

#### E.3.2    AFTER-SALES SERVICE

We included the bad case in Table E4

Table E4: Bad Case in Task After-Sales Service

| Category: Ecom Services - After-Sales Service |
| --- |
| **Question**: Which practice involves refurbishing, remanufacturing, and responsible disposal?
A. Extended Producer Responsibility (EPR)
B. Adopting Sustainable Practices
C. Collaborating with Specialized Logistics Providers
D. Working with Third-Party Recyclers |
| **Correct answer**: B |
| **GPT-5 Response**: C. Collaborating with Specialized Logistics Providers |
| **LLM Judge**: The model response incorrectly identifies option C, while the correct answer is B; thus, the response is incorrect. |
| **Final Score**: 1 |

### E.3.3 LOGISTIC DIALOGUE CLASSIFICATION

We included the bad case in Table E5

Table E5: Bad Case in Task Logistic Dialogue Classification

---

**Category: Ecom Services - Logistic Dialogue Classification**

---

**Question**: You are a helpful assistant that's experienced with customer service of an online e-commerce platform. You provide factual, accurate, professional and valuable responses.Given a conversation between a user and the customer service live agent, determine which of the listed categories is related to the user's question, and response with its ID. (Each category is in presented in the format of "[ID] L1 > L2 > L3", where "ID" is the category ID and "L1 > L2 > L3" is a 3-layer structure, with increasing granularity. The user question's is said to be related to a category if and only if it is related to all L1, L2 and L3.) If none of the category is related to the user's question, then return "NA".

===== Example 1 =====
[Conversation]
[AGENT] Hi gymsportzfitness, thanks for contacting us. Our customer service representative will assist you shortly.
[AGENT] This is Janzel, and I will be assisting you today.
[USER] Hi, orders are not collected yesterday
[USER] Please arrange for driver to pickup the orders by today
[AGENT] I understand that you are concerned about the pickup for your orders. No worries; I will try my best to assist you with this.
[AGENT] May I have 1 order ID, pleasE?
[USER] 1 of order is 240621KKHVBJTP
[AGENT] Thank you for providing the order ID.
[AGENT] Let me check on this real quick. Do you mind if I put you on hold for 2 minutes?
[USER] sure
[AGENT] Thank you so much for the wait.
[AGENT] I know it is important for you to have this issue resolved. I will inform the team to come pick up your item as soon as possible.
[AGENT] No worries; I will be handling this case with high priority, and your item will be picked up. You can count on me to help you with your problem as one of our valued customers.
[AGENT] I will follow up with our fleet team to arrange a pickup. For now, please anticipate the pickup of the parcel within 24 to 48 business hours for a timeframe
[USER] Thank you.
[AGENT] You're most welcome, and I'm happy to know that we were able to assist you.
[AGENT] Is there anything else I can assist you with? It's my pleasure to be of assistance.

[Categories]
- [1423] Logistics > To ship > Rider did not show up/late for pickup
- [8471] Return/Refund > Raising disputes for return/refund requests (NEW flow for selected sellers) > none
- [1849] Seller Operations > Seller Centre / My Shop App > Why is my product deleted/suspended?

[Answer]
1423
===== End of Example 1 =====

===== Example 2 =====
[Conversation]
[AGENT] Hi aromaserapi, thanks for contacting us. Our customer service representative will assist you shortly.
[AGENT] My name is Char, and I will be assisting you today.
[AGENT] Hi, aromaserapi!
[USER] I would like to enquire about the status of my refund
[AGENT] I understand your concern that you want to inquire about your refund status; please let me help you with this.
[AGENT] May I have the order ID of the issue?
[USER] 1799234087724548181
[AGENT] May I confirm that this is the order ID of the issue: 24060466CC4J6K?

[USER] Yes
[AGENT] Please allow me to put you on hold for 2 minutes while I do some quick checking on this matter.
[USER] Okay
[AGENT] Thank you for patiently waiting. Upon checking here. Due to the Lost Parcel by SPX Xpress, you will be receiving reimbursement for the lost parcel in your Seller Balance within 7-14 working days.
[USER] Yes. Its more than 14 working days?
[AGENT] 7 to 14 working days. Yes, but not to worry; it is only a timeframe, and we will come back to you as soon as possible.
[USER] When will it be refunded?
[USER] Today?
[AGENT] Yes.
[USER] Okay.
[USER] Yes, continue
[USER] It will refunded by end of today?
[AGENT] That is correct. In the event that you do not receive the refund within this timeframe, you can contact us again for your concern.
[USER] Okay. I will check
[AGENT] Sure! Is there anything else I can assist you with? It's my pleasure to be of assistance.

[Categories]
- [8617] Logistics > Delivered > Shipping fee related claim
- [1657] General > Seller Penalty Points System > Penalty appeals

[Answer]
NA
===== End of Example 2 =====

Now, think carefully and reply with the related category ID. (Do not provide any additional information.)

[Conversation]
[AGENT] Hi tobyspet, thanks for contacting us. Our customer service representative will assist you shortly.
[USER] 25040206HK4DFQM
[AGENT] Hi, tobyspet. My name is France and I will be assisting you with your request.

How may I assisting you today?
[AGENT] May I know what happen?
[USER] i am Toby
[USER] this request id
[USER] this sku is under promo need buy 2 then get the offer price
[USER] how come now can return one
[USER] totally loss money
[USER] is unfair, i still need pay delivery fee somemore
[AGENT] This is about disagreement with the return refund?
[USER] yes
[AGENT] May I know the detail of the dispute please. Kindly write in one paragraph and I will help you forward to relevant team
[USER] Yes, continue
[USER] this customer order & order id 2503312F3RT7J9, and for this item need get 2 then enjoy the promo price, the customer received this parcel on 1April2025, and after receive he return x1 to us, which is we loss money need pay delivery fee also, it is unfair
[AGENT] Thank you fro clarifying your issue, appricite it.
[AGENT] I will forward to relevant team
[AGENT] Please be informed that I will have this forward to the relevant team for further checking. Kindly anticipate an update within 24-48 hours via our app. Your case ID for reference is 1907705345118556233
[AGENT] I hope it can help you with disputing again
[USER] when can get the result
[USER] and by ws ?
[USER] or email
[USER] or can ask the team call me
[AGENT] It will be 2 working days and we will email you for the update
[USER] ok
[AGENT] We will nudge the team for you to make the process faster
[USER] thanks

1404 [AGENT] We hope the result as you are expected
1405 [AGENT] You're most welcome, I'm glad to help you.
1406 [AGENT] Aside from this, do you have any other concerns?
1407 [USER] the result i expected is compensation
1408 [AGENT] Okay noted, I will inform our relevant team too with that statement
1409 [AGENT] As of the moment, is there anything else I can assist you with? It's my pleasure to be of assistance
1410 [USER] No, End chat

1412 [Categories]
1413 - [3180] Return/Refund > *Dispute > Dispute Evidence - Evidence Requirements
1414 - [2961] Logistics > Order Management > Returned Parcel- Seller Did Not Receive the Parcel
1415 - [1694] Live Agent > Live Agent > Live Agent
1416 - [3152] Return/Refund > RR Issues & Difficulties > Damaged Returned Item - Steps
1417 - [3350] Return/Refund > *Dispute > Disagreement with Outcome
1418 - [983] Logistics > *Order Cancellation > Cancelling Orders that were Shipped
1419 - [2943] Return/Refund > *Dispute > Raising Dispute - Steps
1419 - [1568] Return/Refund > *Dispute > Raising Dispute - About

1420 [Answer]

**Correct answer**: 3350

**gemini-2.5-pro Response**: 2943

**LLM Judge**: The user's main concern is disagreement with the outcome of a return/refund (feeling it's unfair and wishing to dispute), while category 2943 "Raising Dispute - Steps" is about how to raise a dispute, not disagreement with outcome—so model response misses the most precise match.

**Final Score**: 1

### E.3.4    FRAUD RISK ANALYSIS

We included the bad case in Table E6

Table E6: Bad Case in Task Fraud Risk Analysis

| Category: Ecom Services - Fraud Risk Analysis |
|---|
| **Question**: 詐騙類型舉例：
- 導外：廣義的是提到平台以外的方式聯繫或交易，並且由賣家主動要求在平台以外交易
1.包含提供可下單交易的網站或App，如賣貨便
2.包含提供平台以外的付款方式，如Line Pay、超商代碼繳費、無卡分期、貨到付款或其他私下付款方式
3.包含面交/自取相關
4.不包含單純提供通訊軟體資訊
- 廣告：包含
1.其他非電商平台業務訊息(如momo PM招商)
2.募款
3.跟平台無關的廣告（如宣傳集運/物流/廣告公司）

訊息及回答範例：

example 1:
買家: 你好我看貨已經確認送達了請問退款的部分開始處理了嗎
買家: 不好意思請問退款的部分開始處理了嗎
Agent: 賣家您好：訂單編號(******CPFWAD**)訂單已無法於系統申請退貨退款，但買家仍有退貨退款需求。
若您同意退貨退款申請，請採取以下步驟：
匯回款項：
因訂單款項已於完成後撥入平台錢包並提領至賣家銀行帳戶，請協助將款項匯回平台以利退款。 |

銀行名稱：台北富邦銀行-安和分行
銀行代碼：***
分行代碼：****
銀行帳號：************
銀行戶名：*****有限公司台灣分公司

通知平台：
賣家請務必於完成匯款後聯繫平台客服告知以下資訊以執行後續退款作業。
轉出帳號後*碼／轉帳日期及時間／轉帳金額（退款金額）／欲退款之訂單編
號(******CPFWAD**)
賣家: 您好，我這邊向倉庫查詢一下
買家: 我這邊查詢**號上午**點多就已經收到貨了
回答：無嫌疑, 退款

example 2:
賣家: 平台系統規定：
購買【eSIM商品】僅能使用街口支付付款。

如上述因素，無法付款
可Google搜尋晟名科技企業有限公司官網下單
買家: 好的感謝
買家: 已在官網下單成功，感謝您
賣家: eSIM訂單系統已作業，約*-**分鐘會收到email
買家: 好的，感謝您
回答：有嫌疑,導外其他平台下單

參考以上詐騙類型、訊息及回答範例，回答下列對話中，賣家是否有詐騙的嫌疑？
請回答2個值並以","分隔：是否有詐騙嫌疑("有嫌疑"或"無嫌疑"),詐騙類型(若無或不知道回
傳None)
如"無嫌疑,Line", "無嫌疑,正常對話", "無嫌疑,訂金", "無嫌疑,補寄", "有嫌疑,私下匯款"

輸入：
賣家: 打擾了～如果有國際貨物進出口的需求可以聯繫我噢，會比自己進出口便宜很多！！我們公
司主要的業務有：正式報關、包稅清關(不用做商品檢驗、不需要具備該產品進出口資格、不用繳
納進口稅）、集運、快遞進出口、倉儲管理。服務範圍也很廣喔！基本上常聽到的國家都能為您
服務的

重要提示：
- 與遊戲相關的連結、僅要求加入通訊軟體(如Line)、提供優惠價格均不應被視為詐騙
- 若因「補寄商品、退款、退換貨、訂金」的原因要求私下匯款或提供收件資訊，不屬於詐騙範圍
- 賣家提及客戶資料保密的理由可能是正常的商業行為，需綜合考量
- 若賣家在退款過程中要求提供銀行資訊或提供補償金額，無私下交易要求，則不屬於詐騙範圍
- 電話隱碼要求電話並非詐騙
- 提供平台內部的網站不算詐騙
- 銷售帳號或密碼並非詐騙

回答：

**Correct answer**: 有嫌疑,廣告

**Qwen2.5-14b Response**: 無嫌疑,None

**LLM Judge**: The Model Response incorrectly identifies the message as "無嫌疑,None" when it should be
"有嫌疑,廣告" according to the definitions, 賣家在平台販售非平台集運/物流/廣告服務，屬於廣告類
嫌疑。

**Final Score**: 1

## E.4 FAILURE ANALYSIS

Distribution of error types in generative tasks is shown in Figure 3.

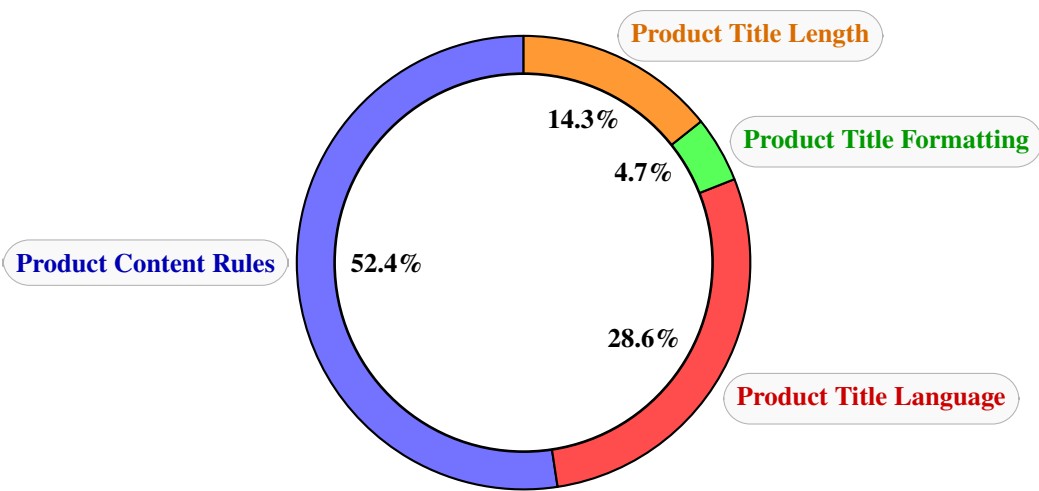

Figure 3: Failure Analysis of Ecom Generative Tasks.

### E.4.1 PRODUCT TITLE CONTENT

Table E7: Product Title Content

| Category: Ecom Generation - Product Title Refinement |
| --- |
| **Question**: **Task:**
Integrate 'value' into [2 MAR (8PM) - 9 MAR | BUY 2 GET 10% OFF] INNISFREE Perfect 9 Intensive Lotion (160ml)|Anti-aging | Firming and Moisture **without modifying origin content**. Output optimized title only.

—

**Processing Pipeline**

**1. Value Preprocessing**
- **Preservation Rules**:
Retain ALL original symbols/emojis/formatting (<...>, 3.5g, 20241014)
Protect multilingual terms in their original form

- **Term Construction**:
**SPF-exclusive combination**:
• **Only** combine 'key_info'+'value' **if** 'key_info="SPF"' (SPF+30→SPF30)
• **All other key_info**: Use raw 'value' directly without prefix
Merge variants/ranges:
• Overlaps: '30,spf30+' → 'SPF30+'
• Ranges: '50ml+100ml' → '50/100ml'
• Deduplicate **per category** (SPF50+SPF50→SPF50)

- **Language Normalization**:
Auto-translate to match adjacent terms' dominant language
Preserve technical codes (PA++++/SPF50) across languages

**2. Insertion Validation**
- **Mandatory Skip Conditions**:
Existing equivalent **in any language** (保湿=Moisturizing)
Logical contradictions (Non-Oily vs Oily) |

Full value/key_info already present (skip if "SPF30" exists when inserting SPF30)

**3. Context-Aware Insertion**
- **Priority Positions**:
1. Adjacent to primary product type (e.g., _Sunscreen_ SPF50)
2. Between high-value term
benefit statement
3. Title end (last resort)

- **Linguistic Alignment**:
Match grammar structure
language flow:
• Insert "保湿" not "Moisturizing" in Chinese-dominant segments
• Mirror adjective/noun order of surrounding text

- **Technical Formatting**:
Cluster related specs (SPF50/PA++++ Protection)
Maintain original casing except technical codes

**4. Absolute Constraints**
- NEVER:
• Remove/modify origin content
• Add conjunctions (with/for/by)
• Repeat existing specs in any language
• Insert 'key_info' text (except SPF technical codes)
• Break protected patterns (<Cosmetics>, 3.5g)

—

**Examples**
Valid:
• 'key_info:SPF + 30' → "Sunscreen SPF30 Cream"
• "Sunscreen SPF50/PA++++ 100ml" (clustered specs)
• 'key_info:SkinType + All skIN' → "Lotion All Skin Face Cream" (no "SkinType" prefix)

Invalid:
• "Moisturizing 保湿Cream" (cross-language duplicate)
• "For SPF50 Milk" (illegal preposition)

—

**Output Format**
Return ONLY the optimized title without explanations.

—

**Input**:
- title: [2 MAR (8PM) - 9 MAR | BUY 2 GET 10% OFF] INNISFREE Perfect 9 Intensive Lotion
(160ml)|Anti-aging | Firming and Moisture
- key_info: Skin Care Benefits
- value: Hydrating

**Correct answer**: [2 MAR (8PM) - 9 MAR | BUY 2 GET 10% OFF] INNISFREE Perfect 9 Intensive
Lotion (160ml)|Anti-aging | Firming and Moisture Hydrating

**GPT-5 Response**: [2 MAR (8PM) - 9 MAR | BUY 2 GET 10% OFF] INNISFREE Perfect 9 Intensive
Lotion (160ml)|Anti-aging | Firming and Moisture

**LLM Judge**: The Model Response failed to insert the value "Hydrating" as required by the instructions,
whereas the Reference Answer did so correctly without modifying the original content.

**Final Score**: 1

E.4.2   PRODUCT TITLE LENGTH

Table E8: Product Title Length

**Category: Ecom Generation - Product Title Generation**

**Question**: You are an e-commerce operation in Indonesia
There is an e-commerce product with the following product information:

- Product title: """BAKSO SAPI PREMIUM SUMBER SELERA KEBON JERUK / BASO PREMIUM SUMBER SELERA ISI 50"""
Sumber Selera
Bakso Sapi Premium Sumber Selera berat 700 gram (isi 50pcs)
Daging sapi yang sangat terasa dan tekstur yang kenyal serta rasa gurih yang begitu enak

Cara penyajian:
Rebus / goreng bakso selama 3-4 menit. Bakso siap disajikan

KUALITAS PREMIUM.
——————————————————————————————————————

Your task is to analyze product information and expand the product title by adding core product features and relevant keywords at the end.
The optimized title must follow the format: [Product title] - [Other core product features]. If there are no [Other core product features], directly return the [Product title].
Requirements for [Product title]:
1. Do not change the current title of the product.
Requirements For [Other core product features]:
Basic Requirements:
1. Use the title as the primary source of facts, with the description as supplementary information. If there is any conflicting information between the title and the description, prioritize the title. Do not include core features that are already present in the product title.
2. Extract only the inherent product features explicitly mentioned in the product information (e.g., type, material, purpose), excluding procedural descriptions (e.g., cooking methods, production methods).
3. Extract only the core product features explicitly mentioned in the product description, and ignore any unmentioned features.
4. Ensure the result accurately reflects the core attributes of the product and does not include any content not explicitly supported in the product information. Avoid semantic misunderstandings by ensuring the result accurately reflects the description.
5. Avoid using words with overlapping meanings to ensure clarity and conciseness.
6. Avoid using words that could be harmful to the sale of the item, such as easy to break, easy to melt and shelf life etc.
7. Avoid using overly broad or generic keywords like ˝Spare Part Mobil Elektronik˝ or ˝Comida Bebida˝.
8. Avoid using keywords that are not related to the product itself, such as services or purchase guarantees or stock information or expiration dates and so on.
9. Must not include promotional text such as best price, discount, top sale, free delivery, ready stock, clearance sale.
10. Do not extract any keywords related to delivery, logistics, or shipping from the description, including ˝instan kirim˝ (fast shipping).
11. Keywords must be in bahasa.
12. Use '-' to connect the original content.
13. Optimized title must not exceed the original title by more than 30 characters.
14. Ensure grammatical correctness and adherence to target language rules.
15. If there are no extractable core features, it is allowed to be left blank.

Additional requirements:
1. Exclude descriptions related to ingredient lists (such as "garam", "gula", "bumbu", etc.).
2. Exclude descriptions related to cooking methods (such as "digoreng","dipanggang", "direbus", etc.).
3. Highlight the core features of the product, while ignoring secondary characteristics such as sweetness.
4. Only include core features (such as main benefits, unique attributes, origin, specifications, etc.), while ignoring secondary characteristics like specific ingredients or minor details.

Secondary Check to ensure the result meets the following requirements:
1. The result accurately reflects the core attributes of the product and does not include any content not explicitly supported in the product information.

2. Please check if the features in the result are all explicitly mentioned core features in the description and retain important features.
3. Does the result accurately reflect the description without semantic misunderstandings?
4. Are there any repeated or redundant words in the product title?
5. Is the optimized result concise enough?
6. Is the result clear and easy to understand?
7. Grammatically correct and follows the rules of the target language.

―――――――――――――――――――――――――――――――――――――――――――

Please only provide the optimized title.

**Correct answer**: BAKSO SAPI PREMIUM SUMBER SELERA KEBON JERUK / BASO PREMIUM SUMBER SELERA ISI 50 - 700 gram

**GPT-4.1 Response**: BAKSO SAPI PREMIUM SUMBER SELERA KEBON JERUK / BASO PREMIUM SUMBER SELERA ISI 50 - 700 gram, daging sapi terasa, tekstur kenyal, rasa gurih

**LLM Judge**: The Model Response includes secondary flavor and texture descriptors ("daging sapi terasa, tekstur kenyal, rasa gurih") which are not considered core features per the task requirements, exceeding both allowed keywords and the 30-character limit, and making the title less concise than needed; Reference Answer meets requirements precisely.

**Final Score**: 1

### E.4.3 PRODUCT TITLE FORMATTING

Table E9: Product Title Formatting

**Category: Ecom Generation - Product Title Refinement**

**Question**: You are a helpful assistant.
您的任务是在不改变原标题内容的前提下，修正商品标题的大小写格式。

标题大小写规范如下：

**基本原则：首字母大写**
即每个单词的首字母大写，但以下情况的单词需要小写，除非它们是标题的第一个单词：
* 冠词：a, an, the
* 介词：in, on, at, by, for, with, to, from, of, over, under 等(这是一个常
见但不完全的列表) * 连词：and, or, but, for, nor, so, yet

**特殊情况：以下词语和情况需要注意**
1. **品牌名称:** 始终按照品牌官方的大小写格式，例如: Apple, Samsung, Sony, ZTE。如果商品有品牌，则品牌格式按照商品的品牌格式
2. **型号和系列名称:** 按照官方型号和系列名称的大小写格式，例如：iPhone 15 Pro Max, Galaxy S23 Ultra, WH-1000XM5。
3. **技术规格和单位:** 技术规格和单位通常为缩写或专有名词，请按照官方格式，例如：GB, mAh, Hz, USB, HDMI, Wi-Fi, Bluetooth。
4. **数字/缩写:** 数字通常保持原样，不进行大小写更改，例如：5G, 4K, 1080p, 3.5mm。

**一些规范的示例：**
1. SanDisk 1TB Extreme PRO Portable SSD - Up to 2000MB/s - USB-C, USB 3.2 Gen 2x2
2. ESR for iPhone 15 Pro Max Case, Compatible with MagSafe, Military-Grade Protection, Yellowing Resistant, Scratch-Resistant Back, Magnetic Phone Case for iPhone 15 Pro Max, Classic Series, Clear

**输出格式：**
1. 严格保持原标题的原始拼写，包括可能的拼写错误或非标准品牌名称。
2. 仅调整大小写格式，不修改单词本身。
3. 直接输出修正后的标题，不添加任何解释或额外文字。

**任务输入：**

商品标题
Softcase Custom Foto Dan Nama Ready Untuk Semua Type Handphone Casing Hp Trand - CS2
商品品牌
Marinitri

**Correct answer**: Marinitri Softcase Custom Foto dan Nama Ready untuk Semua Type Handphone Casing Hp Trand - CS2

**GPT-5 Response**: Softcase Custom Foto Dan Nama Ready Untuk Semua Type Handphone Casing HP Trand - CS2

**LLM Judge**: The Model Response did not apply the brand formatting requirement (missing brand name Marinitri) and incorrectly capitalized connecting words (Dan, Untuk) and "HP" instead of matching the original "Hp"; several title case rules were not followed, so the answer is incorrect.

**Final Score**: 1

### E.4.4 PRODUCT TITLE LANGUAGE

Table E10: Product Title Language

**Category: Ecom Generation - Product Tag Generation**

**Question**: You are an e-commerce expert with rich knowledge of products. Please extract applicable query tags from the following keyword. Return result in English and with JSON format:
"""keyword""":"""xxx""", """tags""":["""tag1""","""tag2""","""tag3"""], do not return any other texts.
keyword: "touca luvinha e meia"

**Correct answer**: {"keyword": "touca luvinha e meia", "query_tags": ["baby clothing", "set", "hat", "mittens", "socks"]}

**Qwen3-30B-A3B Response**: { "keyword": "touca luvinha e meia", "tags": ["touca", "luvinha", "meia", "acessorio", "roupa"] }

**LLM Judge**: The model response provides tags in Portuguese instead of English. It does not follow the language and specificity requirements.

**Final Score**: 1

