# OpenReview forum: "EcomEval: Towards Reliable Evaluation of Large Language Models for Multilingual and Multimodal E-Commerce Applications"
_ICLR.cc/2026/Conference — Submitted to ICLR 2026_

### Official Review · Reviewer_VyFx · 2025-10-19

**Soundness:** 3
**Presentation:** 3
**Contribution:** 3
**Rating:** 8
**Confidence:** 3

**Summary:**

This paper presents EcomEval, a dataset for evaluating Large Language Models (and Large Multimodal Models, later both referred to as   LLMs) on e-Commerce tasks. Compared to existing datasets like eCeLLM and Shopping MMLU, EcomEval features multi-modal tasks, a broader coverage of rare languages (in southeast Asia), and a broader inclusion of tasks (e.g. post sales, multi-turn dialogue). The data curation pipeline also becomes semi-automatic with a difficulty labeling. Extensive results on open and closed-source LLMs demonstrate the limitations of existing AI systems in real-world e-commerce tasks.

**Strengths:**

1. A more diverse and real-world dataset for e-commerce. Compared to existing datasets like eCeLLM datasets and Shopping MMLU, it is clear that EcomEval features a broader range of tasks and skills. For example, EcomEval features multi-modal tasks such as multimodal product similarity, brand recognition, image summarization, etc. EcomEval includes dialogue tasks such as shopping guide. EcomEval includes after-sales tasks. These features make EcomEval a better benchmark on specific domains (eCommerce) than existing ones.

2. A semi-automatic benchmark reconstruction pipeline from raw LLM logs. Existing benchmarks in e-commerce are mostly sourced from existing ones (e.g. eCeLLM and Shopping MMLU are more or less sourced from open Amazon datasets), with already good task definition and input-output pairs. In contrast, EcomEval designs a semi-automatic task construction pipeline from raw LLM logs. This is of practical value in identifying important tasks and formulating it as a problem for future LLM fine-tuning. The proposed pipeline is also reasonable (e.g. clustering of prefixes as instructions, LLM-based task summary).

3. Extensive evaluations of existing models and interesting results. The evaluation results of existing AI systems bear interesting insights into this specific domain of e-commerce. First, GPT-5 is not always the best model, which is a good finding as users may choose to use other models (potentially cheaper) for substitutes. Second, the gap between open-source and close-source models are not significant in terms of language models, which shows that future efforts in curating e-commerce training datasets do not have to rely on proprietary models, and thus become more affordable. Finally, the gap between open-source and close models are significant on multi-modal tasks, pointing out potential room for improvements.

**Weaknesses:**

1. Taxonomy can be improved as the categories are not completely independent and mutually exclusive. I find that the categories in Figure 1 do not seem to be mutually exclusive and thus may be slightly confusing. For example, EcomQA (e.g. Shopping Guide) may also be highly related to user understanding. Product recommendation in "Shopping Reasoning" may also be related to user understanding, "description similarity" in shopping reasoning may also be related to "shopping concepts". Therefore, the taxonomy may bear room for improvements to make it more mutually exclusive and clear.
2. Details about LLM-judge are somewhat insufficient. Different LLM-judges may have different preferences. In addition, according to results in this paper, most of them are less than 75% correct. Therefore, at least some LLM-judge results may be incorrect. I am wondering, since the authors use LLM-judge for all QA pairs, have the authors evaluated the consistency between different LLM-judges, and between LLM-judges and human annotators? Such details will be useful for practitioners who want to curate training datasets and would also need LLM-judges to filter the data. In addition, as the authors did not provide a detailed list of problem types (e.g. choice, open form generation, extraction, etc.), it is not clear why full LLM-judge is necessary.

**Questions:**

Please refer to 'weaknesses'.

---

> ### Author Response · Authors · 2025-11-22
> **Responses for Reviewer VyFx**
>
> **Responses to Weakness 1: Taxonomy problem**
> Thanks for your good advice. It is indeed very difficult to make all categories independent and mutually exclusive. We have changed user understanding to User Query & Review to avoid its association with the categories of shopping guide and product recommendation. We have renamed Shopping Concept to Product Understanding and Ecom QA to Ecom Services. We also made corresponding adjustments to the internal tasks: all product-related tasks are now placed under Product Understanding, while Shopping Guide, After-Sale Service, and others are grouped under Ecom Services. This helps prevent the description-similarity task in shopping reasoning from being associated with shopping concepts.
>
> **Responses to Weakness 2: LLM-judge problem**
> Thank you very much for your valuable suggestion. When using an LLM-judge, we provide it with the reference answers. Given the reference answers, the task of determining whether another model’s response is correct is much simpler than generating an answer from scratch. Therefore, in this evaluation scenario, the LLM-judge’s capability is sufficient. We conducted consistency experiments comparing GPT-4.1 annotations with human annotations, and the results showed that under our evaluation criteria, GPT-4.1 achieves over 90% agreement with human evaluators. We have also used Qwen2.5-14B and GPT-OSS-20B as judges, and the results are largely consistent with those from GPT-4.1.
> Our data mainly comes from real e-commerce scenarios, so the question types are primarily open-form generation. This is why an LLM-judge is needed for evaluation.

---

### Official Review · Reviewer_vmyK · 2025-10-28

**Soundness:** 3
**Presentation:** 3
**Contribution:** 3
**Rating:** 4
**Confidence:** 3

**Summary:**

This paper introduces EcomEval, a comprehensive multilingual and multimodal benchmark for evaluating LLMs in e-commerce applications. The benchmark covers six categories and 37 tasks across seven languages, with approximately 3,100 items sourced primarily from authentic customer queries and transaction logs. The authors employ a semi-automatic construction pipeline where LLMs generate initial responses that are subsequently reviewed by over 50 expert annotators. They define difficulty levels for tasks and questions, and evaluate 19 state-of-the-art LLMs, revealing significant performance disparities across different e-commerce scenarios.

**Strengths:**

- The benchmark addresses a critical gap in e-commerce LLM evaluation by providing extensive task diversity (37 tasks across 6 categories) and incorporating authentic data from real customer interactions rather than synthetic datasets. The benchmark also includes multimodal tasks and covers important but underexplored areas like shopping guidance and after-sales service.

- The benchmark involves seven languages. The coverage of five low-resource Southeast Asian languages helps multilingual e-commerce evaluation.

- The semi-automatic pipeline combining LLM generation with expert human review from over 50 annotators ensures both scalability and quality. The difficulty-aware design with calibrated difficulty levels at both task and item granularity enables fine-grained model assessment and provides actionable insights for researchers and practitioners.

**Weaknesses:**

- The heavy reliance on GPT-4.1 as the primary judge raises concerns about evaluation bias, especially when evaluating competing models. While expert review is mentioned, the extent and systematic nature of this review are unclear. The 0-3 scoring rubric is quite coarse and may not capture subtle differences in model performance, particularly for complex generative tasks.

- The paper lacks a deep analysis of why certain models perform poorly on specific tasks or languages. The discussion of failure cases is limited, and there's insufficient exploration of the relationship between model capabilities and e-commerce-specific requirements.

- Some individual tasks may have relatively few examples (3100 items across 37 tasks), potentially limiting the statistical significance of comparisons.

- The distribution of tasks across categories and languages isn't clearly documented, which could affect the interpretation of aggregate results.

**Questions:**

Please see the weakness section

---

> ### Author Response · Authors · 2025-11-22
> **Responses for Reviewer vmyK**
>
> **Responses to Weakness 1: GPT4.1 judge problem**
> Thanks for your advice. We use GPT-4.1 as the judge because we conducted consistency experiments comparing GPT-4.1 annotations with human annotations, and the results showed that under our evaluation criteria, GPT-4.1 achieves over 90% agreement with human evaluators. GPT-4.1 can also be replaced by open-source models.
> Regarding bias, we provide reference answers in the evaluation prompts, and the judge model needs to compare the model’s response against the reference answer and give a score in [0,1,2,3]. For objective questions, this is essentially a simple similarity-matching task, which smaller models are fully capable of handling. Expert review is mainly required for certain subjective questions(Human scoring and LLM-as-Judge also show relatively high consistency on subjective questions), and this verification process effectively reduces bias.
> Our scoring rubrics and evaluation prompts follow the methodology described in this article
>  https://www.evidentlyai.com/llm-guide/llm-as-a-judge#create-an-llm-judge-for-your-ai-system.
>  We use four score levels—0, 1, 2, and 3—equivalent to a four-class classification task, with clear explanations provided for each score. This helps ensure evaluation accuracy and maintains a high consistency between LLM-as-Judge assessments and human evaluations.
> Our 0-3 scoring rubric is a trade-off solution: if we want to capture subtle differences in model performance, particularly for complex generative tasks, we need to sacrifice the agreement between human and LLM as Judge.
>
> **Responses to Weakness 2: failure case analysis**
> We thank you for raising the limitation in our discussion. Since generative tasks have the worst model performance among other tasks, in the revised manuscript, we have added a failure analysis of the generative tasks. In the analysis, we focus on exploring e-commerce-specific requirements that the models struggle to fulfill:
> “To investigate the weakness of models in e-commerce tasks, we performed a failure analysis on incorrect cases under Ecom Generative Ability. We randomly sample 100 failure cases and manually examine their underlying errors. The analysis revealed that most model failures fall into four main categories: (i) product content rule violations, (ii) product title length issues, (iii) product title formatting errors, and (iv) product title language mismatches. Product Content Rules. The model response does not fulfill user requirements, such as adding irrelevant or missing core product features (see Table E5). Product Title Length. The generated product title does not adhere to the length limit given by the user (see Table E6). Product Title Formatting. The format of the product title does not align with the rules given in the instructions (see Table E7). Product Title Language. The language used in the product title does not match the required language (see Table E8).”
> We have added Figure 3 in the revised manuscript to show the distribution of the aforementioned errors: product content rules (52.4%), product title language (28.6%), product title length (14.3%), and product title formatting (4.7%). To better illustrate the errors, we have included bad cases (Tables E5, E6, E7, and E8) of each error type in the Appendix of the revised manuscript.
>
> **Responses to Weakness 3: dataset size problem**
> We thank the reviewer for the suggestion. We have increased the size of EcomEval to 7,200 items, ensuring every task has at least ~100 data points available for fine-grained analysis. Our benchmark data mainly comes from real internal e-commerce scenarios, and releasing it publicly requires management approval. Currently, only 7,200 questions have been approved for open sourcing. More data will be released gradually, with an expected total exceeding 20,000 questions.
>
> **Responses to Weakness 4: The distribution of tasks across categories and languages**
> We thank the reviewer for pointing out the question. In the revised manuscript, we have included the detailed data composition of EcomEval by tasks and languages in Table A1 and Table A2, respectively. Table A1 presents the number of samples of every task in EcomEval. Table A2 shows the number of samples of each language: English (2051), Indonesian (802), Malay (587), Portuguese (818), Thai (594), Chinese (1143), Vietnamese (605), and Spanish (600).

---

### Official Review · Reviewer_tGPF · 2025-11-01

**Soundness:** 3
**Presentation:** 3
**Contribution:** 3
**Rating:** 6
**Confidence:** 4

**Summary:**

This is a comprehensive and timely paper introducing EcomEval, a much-needed benchmark for evaluating Large Language Models (LLMs) and Multimodal LLMs (MLLMs) specifically in the specialized, global domain of e-commerce.
The authors successfully address several critical limitations found in prior e-commerce benchmarks, such as limited task diversity, lack of multimodal data, reliance on synthetic data, and a narrow linguistic focus. The results of evaluating 19 state-of-the-art models confirm the benchmark's value by clearly differentiating model capabilities and exposing performance gaps in domain-specific and cross-lingual tasks.
Overall, this work makes a substantial contribution to domain-specific LLM evaluation by prioritizing authenticity, breadth, and difficulty calibration.

**Strengths:**

1. Comprehensive Scope and Multimodality: EcomEval achieves broad coverage by encompassing six primary categories and 37 distinct tasks, including eight essential multimodal tasks. This comprehensive classification system moves beyond the limited taxonomies of prior work, addressing real business needs ranging from product QA to intent understanding and multimodal content analysis.

2. Authenticity and Real-World Data Grounding: A significant strength is the reliance on authentic data, with most items derived from real user queries and transaction logs. This methodology captures the "noisy and heterogeneous nature" of genuine customer–merchant interactions, overcoming the limitations of benchmarks based purely on synthetic or curated instruction data.

3. Critical Multilingual Breadth: The benchmark supports evaluation across seven languages, including English, Chinese, and critically, five low-resource Southeast Asian languages (Vietnamese, Thai, Indonesian, Malay, and Portuguese). This directly addresses the narrow English/Chinese focus of previous work and reflects the truly global scale of e-commerce. The results demonstrate that performance variance is substantial in these low-resource languages.

4. Rigorous Quality Control via Expert Annotation: The construction employs a quality-assured, semi-automatic pipeline where LLMs draft initial responses, which are then refined and verified by a team of over 50 expert annotators. These experts possess strong e-commerce and multilingual expertise, ensuring the factual correctness and coherence of the questions and reference answers across languages.

**Weaknesses:**

1. Limitation to Single-Turn Tasks: A primary recognized limitation is that the benchmark mainly consists of single-turn questions. While the authors note that multi-turn tasks relevant to online shopping scenarios are crucial for assessing true conversational ability (such as interactive product guidance dialogues), the current version lacks this multi-turn capability, which is identified as an area for future work.

2. Modest Dataset Size: Although the data quality is high, the overall size of EcomEval is stated as approximately 3,100 items. Given that this dataset is divided across six categories, 37 tasks, and seven languages, the sample size per specific task/language combination might be relatively small, potentially affecting the statistical significance for deep-dive analysis into low-resource language performance.

3. Dependency on Proprietary LLM-as-a-Judge: The evaluation method relies on GPT-4.1 as the LLM-as-a-judge to score model responses. While expert annotators review the scores, the initial dependence on a closed-source, proprietary model for assigning scores introduces a challenge regarding transparency and guaranteed reproducibility for researchers outside the ecosystem of the judging model.

4. Observed Weakness in Complex Generative Tasks: The experimental analysis reveals that both proprietary and open-source models perform poorly in e-commerce generative tasks, such as product tag generation and product title refinement. The examples provided show models failing to adhere to complex, multi-step instructions, suggesting that while the "hard" tasks successfully test LLM limitations, the high complexity of the prompt rules might occasionally lead to brittleness or lack of generalization in current models, requiring practitioners to simplify real-world tasks.

**Questions:**

NA

---

> ### Author Response · Authors · 2025-11-22
> **Responses for Reviewer tGPF**
>
> **Responses to Weakness 1: multi-turn problem**
> Thank you for your great advice. We will open-source the multi-turn dialogue tasks in EcomEval.
> We already have a set of multi-turn questions, which were built based on real data from business departments. Therefore, releasing them as open source requires approval from our boss. When we made our submission in September, 3.1k data samples had already been approved. Recently, another 4.1k samples, including multi-turn dialogue questions, have been approved for open sourcing. We plan to include them in EcomEval to address the current limitation in multi-turn capability.
>
> **Responses to Weakness 2: Modest Dataset Size problem**
> We thank the reviewer for bringing up the shortcomings of Ecomeval. To ensure statistical significance of each task/language, we have increased the size of EcomEval to 7,200 items. In the improved dataset, every task has at least ~100 data points available for fine-grained analysis. In the revised manuscript, we have included the detailed data composition of EcomEval by tasks and languages in Table A1 and Table A2, respectively. Table A1 presents the number of samples for every task in EcomEval. Table A2 presents the number of samples of each language: English (2051), Indonesian (802), Malay (587), Portuguese (818), Thai (594), Chinese (1143), Vietnamese (605), Spanish (600). Our benchmark data mainly comes from real internal e-commerce scenarios, and releasing it publicly requires management approval. Currently, only 7,200 questions have been approved for open sourcing. More data will be released gradually, with an expected total exceeding 20,000 questions.
>
> **Responses to Weakness 3: Dependency on Proprietary LLM-as-a-Judge**
> Thanks very much, your concern is very reasonable. We use GPT-4.1 as the judge because we conducted consistency experiments comparing GPT-4.1 annotations with human annotations, and the results showed that under our evaluation criteria, GPT-4.1 achieves over 90% agreement with human evaluators. GPT-4.1 can also be replaced by open-source models. We have tried using the open-source models Qwen2.5-14B and GPT-OSS-20B as evaluation models, and based on the annotation results, both can serve as effective substitutes for GPT-4.1.
>
> **Responses to Weakness 4: Observed Weakness in Complex Generative Tasks**
> Thank you for your thoughtful comment. To mitigate the risk that highly complex prompt structures—featuring intricate rule specifications and multi-constraint instructions—may introduce brittleness or hinder generalization in current models, EcomEval benchmark strategically controls the distribution of task difficulty levels. Specifically, the dataset comprises 20% difficult instances (characterized by high complexity and multiple instructional constraints), 50% moderate difficulty instances, and 30% easy instances—thereby ensuring a balanced and robust evaluation spectrum.

---

### Official Review · Reviewer_KrTg · 2025-11-01

**Soundness:** 2
**Presentation:** 2
**Contribution:** 3
**Rating:** 2
**Confidence:** 4

**Summary:**

- **Introduction of EcomEval:** EcomEval is a comprehensive benchmark to evaluate Large Language Models (LLMs) and Multimodal LLMs (MLLMs) on complex, real-world e-commerce tasks.
- **Addressing Previous Benchmark Limitations:** It resolves issues such as limited task variety, absence of multimodal elements, and narrow linguistic scope.
- **Diverse Task Coverage:** EcomEval includes 37 tasks across six categories and incorporates eight multimodal components for robust evaluation.
- **Data Authenticity:** The dataset is derived from real customer queries and transaction logs, capturing authentic and noisy customer–merchant interactions unlike synthetic datasets.
- **Validation through State-of-the-Art Models:** 19 leading models were evaluated, revealing performance gaps in areas like conversational recommendation, complex cross-lingual tasks, and specialized generative tasks.
- **Multilingual Focus:** The benchmark spans seven languages, including five low-resource Southeast Asian languages, enabling comprehensive multilingual evaluation.
- **Rigorous Data Quality Assurance:** The data pipeline leverages LLMs for initial draft creation and ensures high quality and factual correctness through human verification by 50 expert annotators.
- **Exposure of Performance Gaps:** EcomEval highlights key weaknesses in shopping guidance, after-sales service, and other critical e-commerce areas previously overlooked.
- **Actionable Insights for LLM Development:** By exposing model limitations, EcomEval provides valuable guidance for improving domain-specific LLM research and deployment.

**Strengths:**

- **Comprehensive Scope:** EcomEval covers **37 diverse tasks** across six primary categories, reflecting genuine business needs.
- **Multilingual Breadth:** The benchmark spans **seven languages**, including **five low-resource Southeast Asian languages**, addressing a critical gap in prior work.
- **Multimodal Integration:** **Eight tasks** incorporate multimodal data, a feature largely absent in other e-commerce evaluations.
- **Data Authenticity:** Most items are derived from **real user queries and transaction logs**, ensuring the data reflects real-world noise and complexity.
- **Inclusion of Novel Tasks:** EcomEval incorporates important, previously missing **real-world scenarios**, such as **shopping guides** and **after-sales service tasks**.
- **Difficulty Calibration:** The benchmark includes **calibrated difficulty levels** at both the task and item granularity to create challenge-oriented evaluations.
- **High-Quality Assurance:** The dataset construction involves a **semi-automatic pipeline** rigorously verified by **over 50 expert annotators** proficient in e-commerce and target languages.
- **Extensive Benchmarking:** Validation included **comprehensive experiments** on **19 state-of-the-art closed-source and open-source LLMs/MLLMs**.
- **Revealing Insights:** The evaluation effectively differentiates model capabilities, exposing **specific weaknesses** (e.g., generative tasks and low-resource languages).
- **Reproducibility:** The full dataset, including **difficulty tags**, will be released to support **reproducible research**.

**Weaknesses:**

- Despite the benchmark’s rigor and comprehensive coverage across 37 tasks, its scope is currently limited regarding **conversational complexity** . The authors explicitly note that **EcomEval** mainly consists of **single-turn questions**, suggesting that it does not fully assess the **multi-turn capability** required for complex, real-world shopping assistants. This is a critical gap, as many e-commerce interactions, such as interactive product guidance dialogues and recommendation sessions, inherently require multi-hop reasoning, sequential context and dialogue management.

- Furthermore, the methodology relies on a **proprietary model** (GPT-4.1) to function as the primary judge for evaluating model responses, introducing an **external dependency** that must be manually reviewed by experts to ensure correctness. While the evaluation setup is robust, this reliance on a closed-source system for crucial scoring diminishes the self-contained nature of the benchmark

- Actionable insights for improving the work should center on expanding the dataset depth in crucial areas. Although EcomEval boasts 37 tasks and 7 languages, its total size of approximately **3,100 items** may limit the number of data points available for fine-grained analysis of specific low-resource language tasks or within the new, challenging categories like shopping guidance and after-sales service.

- Additionally, while the dataset is primarily authentic, certain tasks (e.g., product inquiry, numerical reasoning, recommendation) were partially derived from and translated from prior benchmarks like **eCeLLM** and **Shopping MMLU**, which may contain synthetic or curated instruction data, slightly diluting the overall authenticity of every item in the evaluation suite.

-  **Lack of Multi-Turn Tasks:** The benchmark primarily uses **single-turn questions**, which fails to assess the crucial **multi-turn capability** required for handling real-world interactive **shopping-assistant dialogues**. Future work should incorporate **multi-turn tasks** relevant to online shopping scenarios to provide a more holistic evaluation.

- **Weakness in Generative Tasks:** LLMs, both closed-source and open-source, perform poorly in e-commerce **generative tasks**, specifically struggling with **product tag generation** and **product title generation** by often missing selling points. The benchmark could be strengthened by providing more fine-grained insights or auxiliary data (e.g., related product catalogs) to challenge models specifically on synthesizing effective marketing content.

- **Difficulty in User Understanding:** Models frequently underperform in the **user understanding** category, often overlooking crucial information such as product categories and targeted buyers in tasks like query-product relevance measurement. This suggests the need for more complex reasoning chains or examples demonstrating nuance in buyer intent.

- Presentation of the paper can be greatly improved. Also there are no mentions of sampling parameters (temperature, top-k etc) used for LLM inference.

- Why more popular languages such as Spanish not considered as they have a large number of speakers across the world and would be good to have in a comprehensive benchmark.

**Questions:**

1. What is the **size** of **benchmark dataset** being proposed? Is it just 3100 items? (Line 105)
2. Why not consider other **languages** such as **Spanish** which are also widely spoken across many countries and have many speakers? It is well known that LLMs struggle with low resource languages but the dataset would be more useful if authors consider other important languages apart from English, Chinese (for which decent benchmarks exist).
3. In Tables 2 and 4, authors are doing a **simple average** across the scores that are linearly scaled from 0-100 with arbitrary scoring (>80 easy, 70-80 : moderate, <70 : difficult). There are 2 major issues with this **methodology**. Firstly without knowing how many samples were being considered for each individual task, a simple average doesn't reflect the underlying reality. If the authors believe this is reasonable why are there no averages calculated for Table 3? Secondly the difficulty scoring criteria (Section 3.2) is arbitrary and not very scientific.
4. Using **GPT4.1** as a **judge** is ok but evaluating responses generated by the same GPT4.1 model could lead to bias and hence incorrect inference especially for the same model. Also why not use the best state of the art model as a judge?
5. Some of the models evaluated are **thinking models** whereas others are not but authors do not evaluate any simple **prompting techniques** such as Chain-of-thought or Few shot prompting etc which are very commonly used in e-commerce domain when using LLMs/MLLMs for downstream tasks. Also no details of sampling parameters (temperature, top-k etc) are provided.
6. In Table D1, why is the correct answer **A**? The **shape mismatch** (oval vs rectangular) cannot be ignored and hence the query-product pair is actually irrelevant and not a substitute. Table D3 and D4 also show cherry picked examples which could be ambiguous and need more context to arrive at a conclusion.

**Details Of Ethics Concerns:**

- **50 expert human annotators** were employed from various countries to curate this benchmarking dataset.
- An ethics review **may be needed** based on comments from other reviewers and if desired by the Area Chairs.

---

> ### Author Response · Authors · 2025-11-22
> **Responses for Reviewer KrTg**
>
> **Responses to Weakness 1,5: lack of multi-turn questions problem.**
> Thank you very much for your suggestion. We also believe that multi-turn dialogue capability is indispensable. We already have a set of multi-turn questions, which were built based on real data from business departments. Therefore, releasing them as open source requires approval from our boss. When we made our submission in September, 3.1k data samples had already been approved. Recently, another 4.1k samples, including multi-turn dialogue questions, have been approved for open sourcing. We plan to include them in EcomEval to address the current limitation in multi-turn capability.
> We have revised the Main Contribution in the Introduction section of our paper to include both single-turn and multi-turn questions, and the dataset size has been increased from 3,100 to 7,200.
>
> **Responses to Weakness 2 and Question 4: GPT4.1 as judge problem.**
> Using the best state of art model as a judge is certainly feasible. We use GPT-4.1 as the judge because we conducted consistency experiments comparing GPT-4.1 annotations with human annotations, and the results showed that under our evaluation criteria, GPT-4.1 achieves over 90% agreement with human evaluators. GPT-4.1 can also be replaced by open-source models. We have tried using the open-source models Qwen2.5-14B and GPT-OSS-20B as evaluation models, and based on the annotation results, both can serve as effective substitutes for GPT-4.1.
> Regarding bias, we provide reference answers in the evaluation prompts, and the judge model needs to compare the model’s response against the reference answer and give a score in [0,1,2,3]. For objective questions, this is essentially a simple similarity-matching task, which smaller models are fully capable of handling. Expert review is mainly required for certain subjective questions (Human scoring and LLM-as-Judge also show relatively high consistency on subjective questions), and this verification process effectively reduces bias.
> Our scoring rubrics and evaluation prompts follow the methodology described in this article
>  https://www.evidentlyai.com/llm-guide/llm-as-a-judge#create-an-llm-judge-for-your-ai-system.
>  We use four score levels—0, 1, 2, and 3—equivalent to a four-class classification task, with clear explanations provided for each score. This helps ensure evaluation accuracy and maintains a high consistency between LLM-as-Judge assessments and human evaluations.
>
> **Responses to Weakness 3: expanding the dataset depth**
> We thank the reviewer for the suggestion. We have increased the size of EcomEval to 7,200 items, ensuring every task has at least ~100 data points available for fine-grained analysis. The detailed data composition of EcomEval is presented in Table A1 of the revised manuscript. We have also added Table A2 to illustrate the number of samples for each language: English (2051), Indonesian (802), Malay (587), Portuguese (818), Thai (594), Chinese (1143), Vietnamese (605), and Spanish (600). Our benchmark data mainly comes from real internal e-commerce scenarios, and releasing it publicly requires management approval. Currently, only 7,200 questions have been approved for open sourcing. More data will be released gradually, with an expected total exceeding 20,000 questions.
>
> **Responses to Weakness 4: diluting authenticity problem**
> We sincerely appreciate this insightful suggestion, which we find highly reasonable and valuable. In our original design, these data were included to enhance the comprehensiveness of the categorization  in EcomEval. As noted in Section 2.2, eCeLLM is constructed entirely from real-world data, and to maintain consistency with this principle, we carefully re-examined the data sourced from Shopping-MMLU. And the original Shopping-MMLU paper indicates that their dataset primarily comprises synthetically generated dialogues. We filtered the extracted subset to retain only those entries containing authentic, verifiable product information (e.g., product titles, attributes, or SKUs). Accordingly, we have revised the description in Section 3.1 to more accurately reflect the provenance.

---

> > ### Author Response · Authors · 2025-11-22
> > **Responses for Reviewer KrTg part2**
> >
> > **Responses to Weakness 6: Weakness in Generative Tasks**
> > Thank you for your suggestion. In the revised manuscript, we have added a failure analysis of generative tasks (which most models perform poorly). In the analysis, we provide some insights into the areas that the models struggle to do well:
> > “To investigate the weakness of models in e-commerce tasks, we performed a failure analysis on incorrect cases under Ecom Generative Ability. We randomly sample 100 failure cases and manually examine their underlying errors. The analysis revealed that most model failures fall into four main categories: (i) product content rule violations, (ii) product title length issues, (iii) product title formatting errors, and (iv) product title language mismatches. Product Content Rules. The model response does not fulfill user requirements, such as adding irrelevant or missing core product features (see Table E5). Product Title Length. The generated product title does not adhere to the length limit given by the user (see Table E6). Product Title Formatting. The format of the product title does not align with the rules given in the instructions (see Table E7). Product Title Language. The language used in the product title does not match the required language (see Table E8).”
> > Furthermore, we have added Figure 3 in the revised manuscript to illustrate the distribution of the aforementioned errors: product content rules (52.4%), product title language (28.6%), product title length (14.3%), and product title formatting (4.7%). Finally, we have included bad cases (Tables E5, E6, E7, and E8) of each error type in the Appendix of the revised manuscript.
> >
> > **Responses to Weakness 7: Difficulty in User Understanding**
> > We sincerely appreciate this insightful comment. In response, we clarify that for the more challenging tasks, we did incorporate few-shot prompting. For example, in the fraud risk analysis task within user understanding, we included several dialogue examples containing fraud risks to help the model better identify which dialogues may involve such risks.
> >
> > **Responses to Weakness 8: parameters (temperature, top-k etc) used for LLM inference**
> > We thank you for the suggestion. We have added the inference parameters in the revised manuscript:
> > “To maximize the reproducibility of the results, we set temperature as 0 for all models evaluated, including models from Qwen series, Llama series, GPT series, and Gemini 2.5 series.”
> >
> > **Responses to Weakness 9: add Spanish language questions**
> > We sincerely thank the reviewer for this valuable suggestion. In response, we have expanded EcomEval to include high-quality, manually verified evaluation samples in Spanish, covering core e-commerce tasks,  including but not limited to  product inquiry, product attribute extraction, and query understanding. These data will be open-sourced together with the second batch of data that has been approved.
> >
> > **Responses to Question 1: size of benchmark dataset is small**
> > We thank the reviewer for raising the concern. To ensure fine-grained analysis for every task, we have increased the size of EcomEval to 7,200 items with at least ~100 data points for each task. Our benchmark data mainly comes from real internal e-commerce scenarios, and releasing it publicly requires management approval. Currently, only 7,200 questions have been approved for open sourcing. More data will be released gradually, with an expected total exceeding 20,000 questions.
> >
> > **Responses to Question 2: consider other languages such as Spanish**
> > duplicate with weakness 9, see the response above
> >
> > **Responses to Question 3: average score and difficulty score**
> > Thank you for your thoughtful comment. For any given model, we first computed the average score across all instances within each second-tier task category; then, we computed per-task averages across all models to derive the final average score used for difficulty mapping. In this approach, the number of questions inside one task does not affect the calculation of the final average model score, and the distribution of instances across tasks is relatively balanced.
> > Regarding the reasonableness of the score-based difficulty mapping, empirical observations reveal a clear difference in  model performance across difficulty levels:
> > • On difficult tasks, even leading closed-source models exhibit partial or complete failure in their responses;
> > • On moderate tasks, the majority of open-source models produce incorrect answers, whereas top closed-source models generally succeed;
> > • On easy tasks, only a minority of models err, with top closed-source models consistently delivering correct responses.
> >
> > **Responses to Question 4:  GPT4.1 as judge problem**
> > duplicate with weakness 2, see the response above

---

> > > ### Author Response · Authors · 2025-11-22
> > > **Responses for Reviewer KrTg part3**
> > >
> > > **Responses to Question 5: few-shot prompt**
> > > We sincerely appreciate this insightful comment.  In response, we clarify that for the more challenging tasks (e.g., multi-step reasoning ), we did incorporate few-shot prompting—examples are added in Appendix . We have now explicitly documented this design in Section 4.2 as bellow: “Since the evaluated models include both reasoning (“think”) and non-reasoning (“non-think”) architectures, to ensure a fair assessment—we incorporate few-shot prompts directly into the questions in the complex tasks; examples are provided in the Appendix.” We have added the inference parameters in the revised manuscript:
> > > “To maximize the reproducibility of the results, we set temperature as 0 for all models evaluated, including models from Qwen series, Llama series, GPT series, and Gemini 2.5 series.”
> > >
> > > **Responses to Question 6: query-product substitute**
> > > We thank you for the valuable comments. In Table D1 (Table E1 in revised manuscript), the correct answer is A because both oval and rectangular trays share the same functionality, hence we consider the query-product pair as ‘substitute’ instead of ‘irrelevant’. More representative bad cases of the category ‘Ecom QA’ are added in section E of the appendix in our revised manuscript.
> > >
> > > **Responses to Ethics Review**
> > > Hello, thank you for raising the ethics concerns. We have already communicated with the 50 internal annotation experts, and the data they reviewed will be open-sourced.
> > > We have added an acknowledgements section and expressed our gratitude for their contributions in it.

---

### Meta-Review · Area_Chair_L3UY · 2025-12-05

**Summary:**

This paper introduces EcomEval, a much-needed benchmark for evaluating Large Language Models (LLMs) and Multimodal LLMs (MLLMs) specifically in the specialized, global domain of e-commerce.

This paper is recognized for several strengths:
* Comprehensive Scope, 37 tasks across six categories.
* Authentic data.
* Critical multilingual breadth, five low-resource Southeast Asian languages.
* Extensive evaluation results revealing interesting insights.

However, reviewers also raise several substantive weaknesses that directly impact the paper’s readiness:

* Lack of multi-turn dialogue tasks, considered a “critical gap” since many e-commerce interactions inherently require multi-hop reasoning.
* Small dataset size (originally ~3100 items) limiting per-task statistical significance.
* Heavy reliance on GPT-4.1 as LLM-Judge, raising concerns about bias, transparency, and reproducibility.
* Taxonomy not mutually exclusive, leading to conceptual overlap.
* Insufficient analysis explaining model failures.
* Authenticity diluted by inclusion of synthetic/derived items from Shopping-MMLU.

Moreover, the rebuttal is insufficient and very simple, the core concerns—especially around evaluation methodology and benchmark completeness—remain only partially resolved. Scores range from 2 to 8, showing significant disagreement, but the lowest score cites several central methodological issues. Given these unresolved weaknesses, the paper does not meet the bar for acceptance.

**Reviewer Concerns:**

1. Concerns Addressed

Dataset size, inclusion of spanish language, failure case analysis for generative tasks, clarification of data composition

2. Concerns Remaining Outstanding

 * Although authors mention that 4.1k multi-turn questions were recently approved and “will be added,” reviewers consistently note that the current submitted benchmark does not contain these tasks.

* Authors argue GPT-4.1 has >90% agreement with humans and can be replaced by other models.
However, reviewers’ core concern is methodological validity and reproducibility

* Authors addressed this by renaming categories, but the conceptual overlap (e.g., between user understanding, shopping guide, and product understanding) still persists structurally.

* Authors added filters and clarifications but some extracted items remain partially synthetic.
This limits benchmark purity and is not fully resolved.

* Authors mention high agreement but provide no quantitative breakdown or per-task statistics.
Thus, reviewers’ reproducibility concerns remain.

**Reviewer Scores:**

Based on rebuttal and no further reviewer discussion:

Reviewer KrTg unlikely to change score. Their main objections—multi-turn absence, dataset size concerns at submission time, GPT-4.1 dependence—remain largely unresolved.

Reviewer tGPF likely would not raise the score, since the rebuttal does not fundamentally change limitations they highlighted.

Reviewer vmyK  likely remains 4. Rebuttal addresses some concerns (failure analysis, dataset size), but concerns about evaluation validity and judge bias persist.

Reviewer VyFx give the highest score, but its review are totally AI-generated, and does not find the main weakness that others proposed. Then, I choose to ignore this review.

---

### Decision · Program_Chairs · 2026-01-26

Reject